# TIGHT SECOND-ORDER CERTIFICATES FOR RANDOMIZED SMOOTHING

## ABSTRACT

Randomized smoothing is a popular way of providing robustness guarantees against adversarial attacks: randomly-smoothed functions have a universal Lipschitz-like bound, allowing for robustness certificates to be easily computed. In this work, we show that there also exists a universal curvature-like bound for Gaussian random smoothing: given the exact value and *gradient* of a smoothed function, we compute a lower bound on the distance of a point to its closest adversarial example, called the **S**econd-**o**rder **S**moothing (**SoS**) robustness certificate. In addition to proving the correctness of this novel certificate, we show that SoS certificates are realizable and therefore tight. Interestingly, we show that the maximum achievable benefits, in terms of certified robustness, from using the additional information of the gradient norm are relatively small: because our bounds are tight, this is a fundamental negative result. The gain of SoS certificates further diminishes if we consider the estimation error of the gradient norms, for which we have developed an estimator. We therefore additionally develop a variant of Gaussian smoothing, called *Gaussian dipole smoothing*, which provides similar bounds to randomized smoothing with gradient information, but with much-improved sample efficiency. This allows us to achieve (marginally) improved robustness certificates on high-dimensional datasets such as CIFAR-10 and ImageNet. Code is available at `https://github.com/alevine0/smoothing_second_order`.

## 1 INTRODUCTION

A topic of much recent interest in machine learning has been the design of deep classifiers with provable robustness guarantees. In particular, for an $m$-class classifier $h : \mathbb{R}^d \to [m]$, the $L_2$ *certification problem* for an input $\mathbf{x}$ is to find a radius $\rho$ such that, for all $\delta$ with $\|\delta\|_2 < \rho$, $h(\mathbf{x}) = h(\mathbf{x} + \delta)$. This robustness certificate serves as a lower bound on the magnitude of any adversarial perturbation of the input that can change the classification: therefore, the certificate is a security guarantee against adversarial attacks.

There are many approaches to the certification problem, including exact methods, which compute the precise norm to the decision boundary (Tjeng et al., 2019; Carlini et al., 2017; Huang et al., 2017) as well as methods for which the certificate $\rho$ is merely a lower bound on the distance to the decision boundary (Wong & Kolter, 2018; Gowal et al., 2018; Raghunathan et al., 2018).

One approach that belongs to the latter category is *Lipschitz function approximation*. Recall that a function $f : \mathbb{R}^d \to \mathbb{R}$ is $L$-Lipschitz if, for all $\mathbf{x}, \mathbf{x}'$, $|f(\mathbf{x}) - f(\mathbf{x}')| \le L\|\mathbf{x} - \mathbf{x}'\|_2$. If a classifier is known to be a Lipschitz function, this immediately implies a robustness certificate. In particular, consider a binary classification for simplicity, where we use an $L$-Lipschitz function $f$ as a classifier, using the sign of $f(\mathbf{x})$ as the classification. Then for any input $\mathbf{x}$, we are assured that the classification (i.e, the sign) will remain constant for all $\mathbf{x}'$ within a radius $|f(\mathbf{x})|/L$ of $\mathbf{x}$.

Numerous methods for training Lipschitz neural networks with small, known Lipschitz constants have been proposed. (Fazlyab et al., 2019; Zhang et al., 2019; Anil et al., 2019; Li et al., 2019b) It is desirable that the network be as expressive as possible, while still maintaining the desired Lipschitz property. Anil et al. (2019) in particular demonstrates that their proposed method can universally approximate Lipschitz functions, given sufficient network complexity. However, in practice, for the robust certification problem on large-scale input, randomized smoothing (Cohen et al., 2019) is the

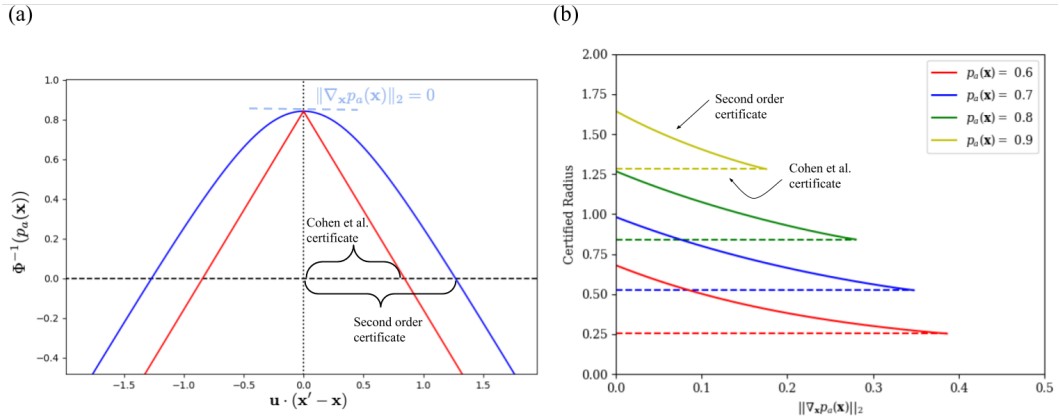

Figure 1: (a) Tight lower bound on the value of a smoothed function at $\mathbf{x}'$ (i.e. $p_a(\mathbf{x}')$) as a function of $\|\mathbf{x}' - \mathbf{x}\|_2$. In this example, $p_a(\mathbf{x}) = 0.8$ and the smoothing standard deviation $\sigma = 1$. The red line shows the lower bound for the function, with no information about the gradient given. The blue line incorporates the additional information that $\|\nabla_{\mathbf{x}} p_a(\mathbf{x})\|_2 = 0$. Note that the axis at $\Phi^{-1}(p_a(\mathbf{x})) = 0$ corresponds to $p_a(\mathbf{x}) = 0.5$, the decision boundary for a binary classifier. (b) Tight robustness certificates for a randomized-smoothed classifier, given the top-class value $p_a(\mathbf{x})$ and the gradient norm $\|\nabla_{\mathbf{x}} p_a(\mathbf{x})\|_2$. The dashed lines show the certificates given $p_a(\mathbf{x})$ alone. Note that the maximum possible gradient for a smoothed classifier depends on $p_a(\mathbf{x})$ (see Equation 1).

current state-of-the-art method. The key observation of randomized smoothing (as formalized by (Salman et al., 2019; Levine et al., 2019)) is that, for any arbitrary *base classifier* function $f : \mathbb{R}^d \to [0, 1]$, the function

$$\mathbf{x} \to \Phi^{-1}(p_a) \quad \text{where} \quad p_a(\mathbf{x}) := \mathop{\mathbb{E}}_{\epsilon \sim \mathcal{N}(0, \sigma^2 I)} f(\mathbf{x} + \epsilon) \tag{1}$$

is $(1/\sigma)$-Lipschitz, where $\mathcal{N}(0, \sigma^2 I)$ is a $d$-dimensional isometric Gaussian distribution with variance $\sigma^2$ and $\Phi^{-1}$ is the inverse normal CDF function. As a result, given the smoothed classifier value $p_a(\mathbf{x})$ at $\mathbf{x}$, one can calculate the certified radius $\rho(\mathbf{x}) = \sigma\Phi^{-1}(p_a(\mathbf{x}))$ in which $p_a(\mathbf{x}) \geq 0.5$ (i.e., $\Phi^{-1}(p_a(\mathbf{x})) \geq 0$). This means that we can use $p_a(\mathbf{x}) \in \mathbb{R}^d \to [0, 1]$ as a robust binary classifier (with one class assignment if $p_a(\mathbf{x}) \geq 0.5$, and the other if $p_a(\mathbf{x}) < 0.5$). Cohen et al. (2019) shows that this is a *tight* certificate result for a classifier smoothed with Gaussian noise: given the value of $p_a(\mathbf{x})$, there exists a base classifier function $f$ such that, if $p_a$ is the Gaussian-smoothed version of $f$, then there exists an $\mathbf{x}'$ with $\|\mathbf{x} - \mathbf{x}'\|_2 = \rho$ such that $p_a(\mathbf{x}') = 0.5$. In other words, the certificate provided by (Cohen et al., 2019) is the largest possible certificate for Gaussian smoothing, given only the value of $p_a(\mathbf{x})$. Previous results (Li et al., 2019a; Lecuyer et al., 2019) provided looser bounds for Gaussian smoothing.

Singla & Feizi (2020) have recently shown, for shallow neural networks, that, rather than globally bounding the (first-order) Lipschitz constant of the network, it is possible to achieve larger robustness certificates by instead globally bounding the Lipschitz constant of the *gradient* of the network. This second-order, curvature-based method takes advantage of the fact that the gradient at $\mathbf{x}$ can be computed easily via back-propagation, so certificates can make use of both $f(\mathbf{x})$ and $\nabla_{\mathbf{x}} f(\mathbf{x})$.

This leads to a question: can we also use the gradient of a smoothed classifier $\nabla_{\mathbf{x}} p_a(\mathbf{x})$ to improve smoothing-based certificates? *In this work, we show that there is a universal curvature-like bound for all randomly-smoothed classifiers.* Therefore, given $p_a(\mathbf{x})$ and $\nabla_{\mathbf{x}} p_a(\mathbf{x})$, we can compute larger certificates than is possible using the value of $p_a(\mathbf{x})$ alone. Moreover, our bound is tight in that, given only the pair $(p_a(\mathbf{x}), \nabla_{\mathbf{x}} p_a(\mathbf{x}))$, the certificate we provide is the largest possible certificate for Gaussian smoothing. We call our certificates "Second-order Smoothing" (SoS) certificates. As shown in Figure 1, the smoothing-based certificates which we can achieve using second-order smoothing represent relatively modest improvements compared to the first-order bounds. This is a meaningful negative result, given the tightness of our bounds, and is therefore useful in guiding (or limiting) future research into higher-order smoothing certificates. Additionally, this result shows that

randomized smoothing (or, specifically, functions in the form of Equation 1) can *not* be used to universally approximate Lipschitz functions: all randomly smoothed functions will have the additional curvature constraint described in this work.

If the base classifier $f$ is a neural network, computing the expectation in Equation 1 analytically is not tractable. Therefore it is standard (Lecuyer et al., 2019; Cohen et al., 2019; Salman et al., 2019) to estimate this expectation using $N$ random samples, and bound the expectation probabilistically. The certificate is then as a high-probability, rather than exact, result, using the estimated lower bound of $p_a(\mathbf{x})$. In Section 3.1, we discuss empirical estimation of the gradient norm of a smoothed classifier for second-order certification, and develop an estimator for this quantity, in which the number of samples required to estimate the gradient scales linearly with the dimensionality $d$ of the input.[1] In order to overcome this, in Section 4, we develop a modified form of Gaussian randomized smoothing, Gausian Dipole Smoothing, which allows for a *dipole certificate*, related to the second-order certificate, to be computed. Unlike the second-order certificate, however, the dipole certificate has no explicit dependence of dimensionality in its estimation, and therefore can practically scale to real-world high-dimensional datasets.

## 2 PRELIMINARIES, ASSUMPTIONS AND NOTATION

We use $f(\mathbf{x})$ to represent a generic scalar-valued "base" function to be smoothed. In general, we assume $f \in \mathbb{R}^d \to [0, 1]$. However, for empirical estimation results (Theorem 3), we assume that $f$ is a "hard" base classifier: $f \in \mathbb{R}^d \to \{0, 1\}$. This will be made clear in context. The *smoothed* version of $f$ is notated as $p_a \in \mathbb{R}^d \to [0, 1]$, defined as in equation 1.

Recall that $\Phi$ is the normal CDF function and $\Phi'$ is the normal PDF function. In randomized smoothing for multi-class problems, the base classifier is typically a vector-valued function $\mathbf{f} \in \mathbb{R}^d \to \{0, 1\}^m, \sum_c \mathbf{f}_c(\mathbf{x}) = 1$, where $m$ is the number of classes. The final classification returned by the smoothed classifier is then given by $a := \arg\max_c \mathbb{E}_\epsilon \mathbf{f}_c(\mathbf{x} + \epsilon)$. However, in most prominent implementations (Cohen et al., 2019; Salman et al., 2019), certificates are computed using only the smoothed value for the estimated *top* class $a$, where $a$ is estimated using a small number $N_0$ of initial random samples, before the final value of $p_a(\mathbf{x})$ is computed using $N$ samples. The certificate then determines the radius in which $p_a(\mathbf{x}')$ will remain above $0.5$: this guarantees that $a$ will remain the top class, regardless of the other logits. While some works (Lecuyer et al., 2019; Feng et al., 2020) independently estimate each smoothed logit, this incurs additional estimation error as the number of classes increases. In this work, we assume that only estimates for the top-class smoothed logit $p_a(\mathbf{x})$ and its gradient $\nabla_\mathbf{x} p_a(\mathbf{x})$ are available (although we briefly discuss the case with more estimated logits in Section 3.2). When discussing empirical estimation, we use $\eta$ as the accepted probability of failure of an estimation method.

## 3 SECOND-ORDER SMOOTHING CERTIFICATE

We now state our main second-order robustness certificate result:

**Theorem 1.** *For all* $\mathbf{x}, \mathbf{x}'$ *with* $\|\mathbf{x} - \mathbf{x}'\|_2 < \rho$, *and for all* $f : \mathbb{R}^d \to [0, 1]$,

$$p_a(\mathbf{x}') \geq \Phi\left(\Phi^{-1}(a' + p_a(\mathbf{x})) - \frac{\rho}{\sigma}\right) - \Phi\left(\Phi^{-1}(a') - \frac{\rho}{\sigma}\right) \tag{2}$$

*where* $a'$ *is the (unique) solution to*

$$\Phi'(\Phi^{-1}(a')) - \Phi'(\Phi^{-1}(a' + p_a(\mathbf{x}))) = -\sigma\|\nabla_\mathbf{x} p_a(\mathbf{x})\|_2. \tag{3}$$

*Further, for all pairs* $(p_a(\mathbf{x}), \|\nabla_\mathbf{x} p_a(\mathbf{x})\|_2)$ *which are possible, there exists a base classifier* $f$ *and an adversarial point* $\mathbf{x}'$ *such that Equation 2 is an equality. This implies that our certificate is realizable, and therefore tight.*

Note that the right-hand side of Equation 2 is monotonically decreasing with $\rho$: we can then compute a robustness certificate by simply setting $p_a(\mathbf{x}') = 0.5$ and solving for the certified radius $\rho$. Also,

---

[1]In a concurrent work initially distributed after the submission of this work, Mohapatra et al. (2020) have proposed an identical second-order smoothing certificate, along with a tighter empirical estimator for the gradient norm. In this estimator, the number of samples required scales with $\sqrt{d}$.

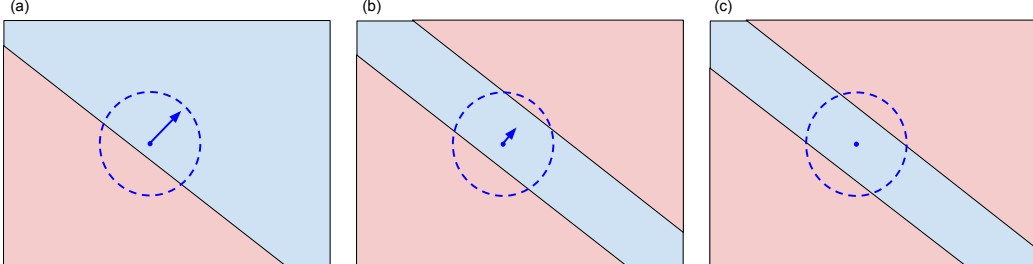

Figure 3: Worst case base classifiers for second-order smoothing for the same value of $p_a(\mathbf{x})$ at different values of $\|\nabla_{\mathbf{x}} p_a(\mathbf{x})\|_2$. The base classifier is $f = 1$ in the blue regions and $f = 0$ in the red regions. The point $\mathbf{x}$ is shown as a blue dot, with the Gaussian sampled region used for calculating $p_a(\mathbf{x})$ is approximately shown as a dashed blue circle. $\nabla_{\mathbf{x}} p_a(\mathbf{x})$ is shown as a blue arrow. (a) The gradient takes its maximum possible value: $\|\nabla_{\mathbf{x}} p_a(\mathbf{x})\|_2 = \sigma^{-1} \Phi'(\Phi^{-1}(p_a(\mathbf{x})))$. (b) The gradient has an intermediate value: $0 < \|\nabla_{\mathbf{x}} p_a(\mathbf{x})\|_2 < \sigma^{-1} \Phi'(\Phi^{-1}(p_a(\mathbf{x})))$. (c) The gradient is zero: $\|\nabla_{\mathbf{x}} p_a(\mathbf{x})\|_2 = 0$.

$a'$ can be computed easily, because the left-hand side of Equation 3 is monotonic in $a'$. Evaluated certificate values are shown in Figure 1-b, and compared with first-order certificates.

All proofs are presented in Appendix A. Like in Cohen et al. (2019), we proceed by constructing the *worst-case* base classifier $f$ given $p_a(\mathbf{x})$ and $\|\nabla_{\mathbf{x}} p_a(\mathbf{x})\|_2$. This is the base classifier $f$ which creates an adversarial point to the smoothed classifier as close as possible to $\mathbf{x}$, given the constraints that $p_a(\mathbf{x})$ and $\|\nabla p_a(\mathbf{x})\|_2$ are equal to their reported values. In Cohen et al. (2019), given only $p_a(\mathbf{x})$, this is simply a linear classifier. With the gradient norm, the worst case is that $\mathbf{x}$ lies in a region with class $a$ which is a slice between *two* linear decision boundaries, both perpendicular to $\nabla p_a(\mathbf{x})$. See Figure 3. Note that, by isometry and because $\nabla p_a(\mathbf{x})$ is the only vector information we have, there is no benefit in certified radius to having the direction of $\nabla p_a(\mathbf{x})$: the norm is sufficient. In the case of a linear classifier the gradient takes its maximum possible value: $\|\nabla_{\mathbf{x}} p_a(\mathbf{x})\|_2 = \sigma^{-1} \Phi'(\Phi^{-1}(p_a(\mathbf{x})))$. This case is shown in Figure 3-a: if the gradient norm is equal to this value, the second-order certificate is identical to the first-order certificate (Cohen et al., 2019). However, if the gradient norm is smaller, then we *cannot* be in this

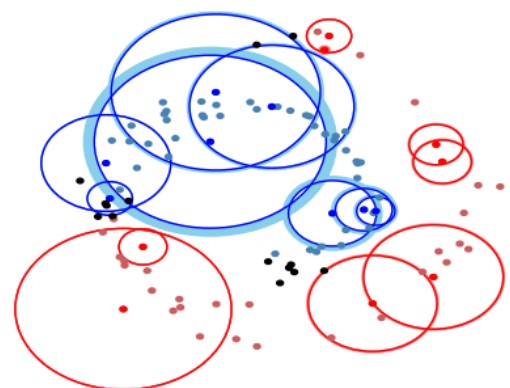

Figure 2: Comparison of second-order smoothing certificates to standard Gaussian smoothing certificates on a selection of points from the Swiss Roll dataset. Correctly labeled points with (second-order) certificates are shown in light red and blue, and points with incorrect label or no certificate are in black. For a selection of points, shown in red/blue, the first-order certified radii shown are as red/blue rings. Increases to certified radii due to second-order smoothing shown are as light blue (light red, absent) rings around certificate radii. For both experiments, $N = 10^8$, and $\eta = 0.001$.

worst-case linear-classifier scenario. Instead, the new "worst case" is constructed by introducing a second "wrong class" region opposite to the direction of the adversarial point (Figure 3-b). In the extreme case (Figure 3-c) where the gradient norm is zero, this is accomplished by balancing two adversarial regions in a "sandwich" around $\mathbf{x}$.

This "sandwich" configuration reveals the relative weakness of gradient information in improving robustness certificates: having zero gradient does *not* require that the adversarial regions be evenly distributed around $\mathbf{x}$. Rather, it is sufficient to distribute the adversarial probability mass $1 - p_a(\mathbf{x})$ into just two adversarial regions. Therefore, the certified radius, even in this most extreme case, is

similar to the Cohen et al. (2019) certificate in the case with half as much adversarial probability mass (the first-order certificate for $p_a(\mathbf{x}) := (1 + p_a(\mathbf{x}))/2$). This can be seen in Figure 1-b: note that at $p_a(\mathbf{x}) = 0.6$, if the gradient norm is known to be zero, the certificate is slightly below the certificate for $p_a(\mathbf{x}) = 0.8$ with no gradient information. The second-order certificate when $(p_a(\mathbf{x}) = 0.6, \|\nabla_{\mathbf{x}} p_a(\mathbf{x})\|_2 = 0)$ is in fact slightly below the first-order certificate for $p_a(\mathbf{x}) = 0.8$, because the Gaussian noise samples throughout all of space, so the smoothed classifier decision boundary is slightly affected by the adversarial region in the opposite direction of $\mathbf{x}$.

Because we can explicitly construct "worst-case" classifiers which represent the equality case of Equation 2, our certificates are known to be tight: the reported certified radii are the largest possible certificates, if only $p_a(\mathbf{x})$ and $\|\nabla p_a(\mathbf{x})\|_2$ are known.

In Figure 2, we show how our second-order certificate behaves on a simple, two-dimensional, non-linearly separable dataset, the classic Swiss Roll. The increases are marginal, mostly because the certificates using standard randomized smoothing are already fairly tight. On these data, the certified radii for the two classes are nearly touching in many places along the decision boundary. However, for the blue class, which is surrounded on multiple sides by the red class, there are noticeable increases in the certified radius. This is especially true for points near the center of the blue class, which are at the "top of the hill" of the blue class probability, and therefore have smaller gradient.

### 3.1 GRADIENT NORM ESTIMATION

In order to use the second-order certificate in practice, we must first bound, with high-probability, the gradient norm $\|\nabla_{\mathbf{x}} p_a(\mathbf{x})\|_2$ using samples from the base classifier $f$. Because Theorem 1 provides certificates that are strictly decreasing with $\|\nabla_{\mathbf{x}} p_a(\mathbf{x})\|_2$, it is only necessary to lower bound $\|\nabla_{\mathbf{x}} p_a(\mathbf{x})\|_2$ with high probability.

Salman et al. (2019) suggest two ways of approximate the gradient vector $\nabla_{\mathbf{x}} p_a(\mathbf{x})$ itself, both based on the following important observation:

$$\nabla_{\mathbf{x}} p_a(\mathbf{x}) = \mathop{\mathbb{E}}_{\epsilon \sim \mathcal{N}(0,\sigma^2 I)} [\nabla_{\mathbf{x}} f(\mathbf{x} + \epsilon)] = \mathop{\mathbb{E}}_{\epsilon \sim \mathcal{N}(0,\sigma^2 I)} [\epsilon f(\mathbf{x} + \epsilon)]/\sigma^2 \tag{4}$$

These two methods are:

1. At each sampled point, one can measure the gradient of $f$ using back-propagation, and take the mean vector of these estimates.

2. At each sampled point, one can multiply $f(\mathbf{x} + \epsilon)$ by the noise vector $\epsilon$, and take the mean vector of these estimates.

Note, however, that Salman et al. (2019) does not provide statistical bounds on these estimates: for our certificate application, we must do so. While we ultimately use an approach based on method 2, we will first briefly discuss method 1. The major obstacle to using method 1 is that it requires that the base classifier $f$ itself to be a Lipschitz function, with a small Lipschitz constant. This can be understood from Markov's inequality. For example, consider the value of some component $z(\mathbf{x}) := \mathbf{u} \cdot \nabla f(\mathbf{x})$, where $\mathbf{u}$ is an arbitrary vector. Suppose N samples are taken, but that $z$ is distributed such that:

$$z(\mathbf{x} + \epsilon) = \begin{cases} 0 & \text{with probability } 1 - \frac{1}{2N} \\ 2N & \text{with probability } \frac{1}{2N} \end{cases} \tag{5}$$

This would be the case if $f$ is a function that approximates a step function from 0 to 1, with a small buffer region of very high slope, for example. Note that the probability that *any* of the $N$ samples measures the nonzero gradient component is $< 0.5$, but the expected value of this component is in fact $1.0$. This example shows that, in order to accurately estimate the gradient with high probability, the number of samples used must *at least* scale linearly with the *maximum* possible value of the gradient norm for $f$. For un-restricted deep neural networks, Lipschitz constants are NP-hard to compute, and upper bounds on them are typically very large (Virmaux & Scaman, 2018). Of course, we could use Lipschitz-constrained networks as described in Section 1 for the base classifier, but this would defeat the purpose of using randomized smoothing in the first place. Moreover, in standard "hard" randomized smoothing as typically implemented (Cohen et al., 2019; Salman et al., 2019), the range of $f$ is $\{0, 1\}$, so $f$ is non-differentiable: therefore, this back-propagation method can not be used at all.

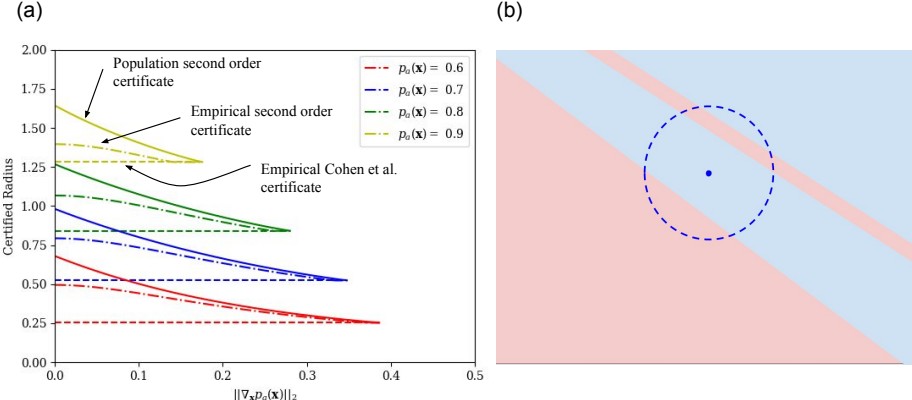

Figure 4: (a) Empirical second-order smoothing certificates, with $d = 49$ (corresponding to $7 \times 7$ MNIST experiments), $N = 10^8$, and $\eta = .001$ (b) Worst case classifier for dipole smoothing.

We therefore use method 2. In particular, we reject the naive approach of estimating each component independently, taking a union bound, and the taking the norm: not only would the error in the norm-squared scale with $d$ as the error from each component accumulates, but there would be an additional dependence on $d$ from the union bound: each component would have to be bounded with failure probability $\eta/d$, where $\eta$ is the total failure probability for measuring the gradient norm. Note that this issue will also be encountered in method 1 above, but in that case, a loose upper bound could at least be achieved without this dependency using Jensen's inequality (the mean of the norms of the gradient is larger than the norm of the mean).

Instead, we estimate the norm-squared of the mean using a single, unbiased estimator. Note that:

$$
\begin{aligned}
\|\nabla_{\mathbf{x}} \mathbb{E}_{\epsilon}[f(\mathbf{x} + \epsilon)]\|_2^2 = \sigma^{-4} \mathbb{E}_{\epsilon}[\epsilon f(\mathbf{x} + \epsilon)] \cdot \mathbb{E}_{\epsilon}[\epsilon f(\mathbf{x} + \epsilon)] = \\
\sigma^{-4} \mathbb{E}_{\epsilon}[\epsilon f(\mathbf{x} + \epsilon)] \cdot \mathbb{E}_{\epsilon'}[\epsilon' f(\mathbf{x} + \epsilon')] = \\
\sigma^{-4} \mathbb{E}_{\epsilon, \epsilon'}[(\epsilon f(\mathbf{x} + \epsilon)) \cdot (\epsilon' f(\mathbf{x} + \epsilon'))]
\end{aligned}
\tag{6}
$$

In other words, we can estimate the norm-squared of the mean by taking pairs of smoothing samples, and taking the dot product of the noise vectors times the product of the sampled values. We show that this is a subexponential random variable (see Appendix), which gives us an asymptotically linear scaling of $N$ with $d$:

**Theorem 2.** *Let $V := \mathbb{E}_{\epsilon, \epsilon'}[(\epsilon f(\mathbf{x} + \epsilon)) \cdot (\epsilon' f(\mathbf{x} + \epsilon'))]$, and $\tilde{V}$ be its empirical estimate. If $n$ pairs of samples $(= N/2)$ are used to estimate $V$, then, with probability at most $\eta$, $\mathbb{E}[V] - \tilde{V} \geq t$, where:*

$$
t = \begin{cases} 4\sigma^2 \sqrt{-\frac{d}{n} \ln(\eta)} & \text{if } -2\ln(\eta) \leq dn \\ -\frac{4\sqrt{2}\sigma^2}{n} \ln(\eta) & \text{if } -2\ln(\eta) > dn \end{cases}
\tag{7}
$$

Note that in practice, we can use the same samples to estimate $\|\nabla_{\mathbf{x}} p_a(\mathbf{x})\|_2$ as are used to estimate $p_a(\mathbf{x})$. However, this requires reducing the failure probability of each estimate to $\eta' = \eta/2$, in order to use a union bound. This means that, if $N$ is small (or $d$ large), second-order smoothing can in fact give worse certificates than standard smoothing, because the benefit of a (loose, for $N$ small) estimate of the gradient is less significant than the negative effect of reducing the estimate of $p_a(\mathbf{x})$. As shown in Figure 4-a, even for very large $N$ and relatively small dimension, the empirical estimation significantly reduces the radii of certificates which can be calculated. See Section 5 for experimental results.

## 3.2 Upper-bound and multi-class certificates

We can easily convert Theorem 1 into a tight upper bound on $p_a(\mathbf{x}')$ by simply evaluating it for $f' = 1 - f$ (and therefore $p_a' = 1 - p_a$). If estimates and gradients are available for multiple classes,

it would then be possible to achieve an even larger certificate, by setting the lower bound of the top logit equal to the upper bounds of each of the other logits. Note, however, that unlike first-order smoothing works (Lecuyer et al., 2019; Feng et al., 2020) which use this approach, it is not sufficient to compare against just the "'runner-up" class, because other logits may have less restrictive upper-bounds due to having larger gradients. As discussed above, gradient norm estimation can be computationally expensive, so gradient estimation for many classes may not be feasible. Also, note that while this approach would produce larger, correct certificates, we do *not* claim that these would be *tight* certificates given the value and gradient information for *all* classes: the "worst case" constructions we describe above for a single logit might not be simultaneously construct-able for multiple logits.

## 4  DIPOLE SMOOTHING

For large-scale image datasets, the dependence on $d$ in Theorem 2 can create statistical barriers. However, the general approach of second-order smoothing, especially using the discrete estimation method (method 2) described above, has an interesting interpretation: rather than using simply the mean of $f(\mathbf{x} + \epsilon)$, we are also using the *geometrical* distribution of the values of $f(\mathbf{x} + \epsilon)$ in space to compute a larger certified bound. In particular, if we can show that points which are adversarial for the base classifier (points with $f(\mathbf{x} + \epsilon) = 0$) are *dispersed*, then this will imply larger certificates, because it makes it impossible for a perturbation in a single direction to move $\mathbf{x}$ towards the adversarial region. Second-order smoothing, above, is merely an example of this.

We therefore introduce *Gaussian Dipole smoothing*. This is a method which, like second-order smoothing, also harnesses the geometrical distribution of the values of $f(\mathbf{x})$ to improve certificates. However, unlike second-order smoothing, there is no explicit dependence on $d$ in the empirical dipole smoothing bound. In this method, when we sample $f(\mathbf{x} + \epsilon)$ when estimating $p_a(\mathbf{x})$, we also sample $f(\mathbf{x} - \epsilon)$. This allows us to compute two quantities:

$$
\begin{aligned}
C^S &:= \mathbb{E}_\epsilon[f(\mathbf{x} + \epsilon)f(\mathbf{x} - \epsilon)] \\
C^N &:= \mathbb{E}_\epsilon[f(\mathbf{x} + \epsilon) - f(\mathbf{x} + \epsilon)f(\mathbf{x} - \epsilon)]
\end{aligned}
\tag{8}
$$

The certificate we can calculate is then as follows:

**Theorem 3.** *For all $\boldsymbol{x}, \boldsymbol{x}'$ with $\|\boldsymbol{x} - \boldsymbol{x}'\|_2 < \rho$, and for all $f : \mathbb{R}^d \to [0, 1]$,*

$$
p_a(\mathbf{x}') \geq \Phi\left(\Phi^{-1}(C^N) - \frac{\rho}{\sigma}\right) + \Phi\left(\Phi^{-1}(\frac{1+C^S}{2}) - \frac{\rho}{\sigma}\right) - \Phi\left(\Phi^{-1}(\frac{1-C^S}{2}) - \frac{\rho}{\sigma}\right)
\tag{9}
$$

We also compute this bound by constructing the worst possible classifier. In this case, the trick is that, if two adversarial sampled points are opposite one another (i.e., $f(\mathbf{x} + \epsilon) = f(\mathbf{x} - \epsilon) = 0$) then they cannot both contribute to the same adversarial "direction". In the worst case, the "reflected" adversarial points form a plane opposite the base classifier decision boundary (See Figure 4-b). In the extreme case where $C^N = 0$, the "worst case" classifier is the same as for second-order smoothing.

Experimentally, we simply need to lower-bound both $C^S$ and $C^N$ from samples. This reduces the precision of our estimates, for two reasons: we have half as many *independent* samples for the same number of evaluations we must perform, and we are bounding two quantities, which requires halving the error probability for each. However, unlike second-order smoothing, there is no dependence on $d$: this allows for practical certificates of real-world datasets.

## 5  EXPERIMENTS

Experimental results are presented in Figures 5 and 6, with further results in Appendix B. Because both dipole and second-order certificates reduce the precision with which empirical quantities needed for certification can be estimated, but both provide strictly larger certificates at the population level, the key question becomes at what number of samples $N$ does each higher-order method

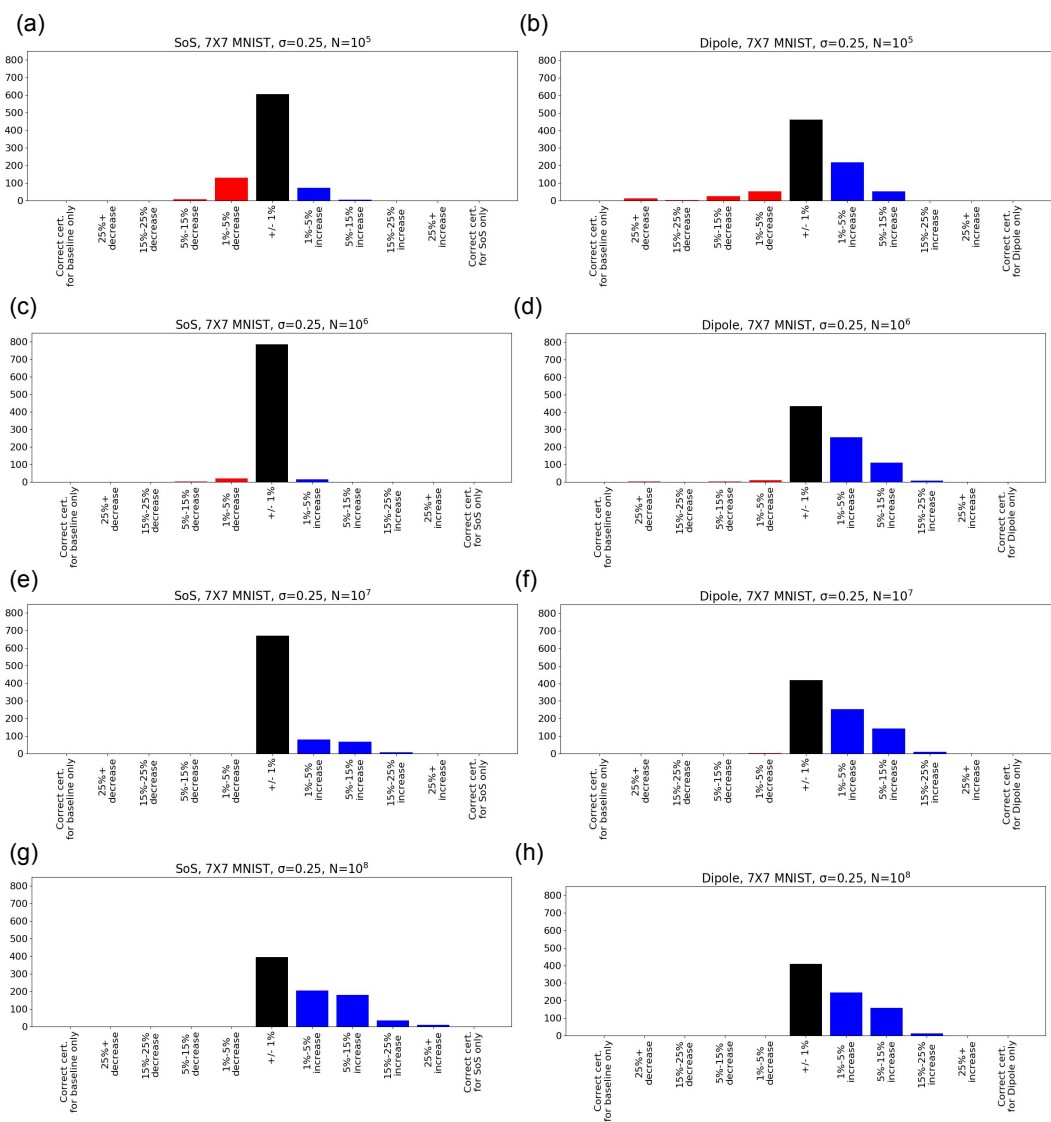

Figure 5: Experiments on $7 \times 7$ MNIST. Reported is the distribution of the improvement (or reduction) of higher-order certificates from certificates computed using standard (first-order) randomized smoothing, for each tested image. For all, $\sigma = 0.25$. For (a, c, e, g), Second-order Smoothing is used. For (b, d, f, h), Gaussian dipole smoothing is used. For (a, b), $N = 10^5$. For (c, d), $N = 10^6$. For (e, f), $N = 10^7$. For (g, h), $N = 10^8$.

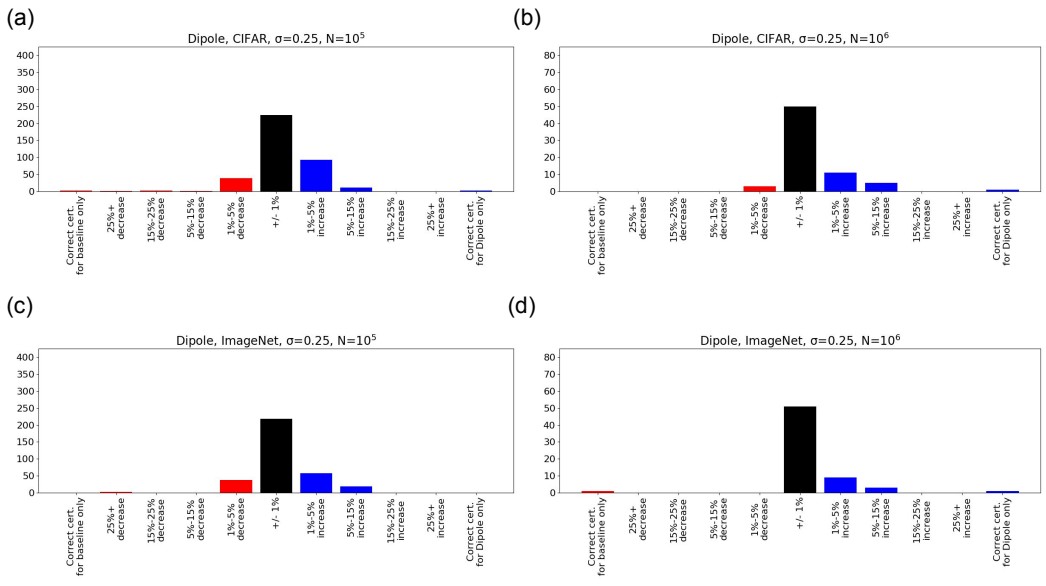

Figure 6: Dipole smoothing experiments, with $\sigma = 0.25$, on CIFAR-10 (a,b) and ImageNet (c,d). For (a,c), $N = 10^5$. For (b,d), $N = 10^6$.

become beneficial. Note that in the figures, we are comparing the new methods to standard smoothing, using the *same $N$* for standard smoothing as for the new method. Due to the poor scaling of second-order certificates with dimension, we tested second-order smoothing on a low-dimensional dataset, $7 \times 7$ MNIST. However, significant increases to certificates were not seen until $N = 10^7$ even on this dataset. By contrast, dipole smoothing is beneficial for many images even when smaller numbers of smoothing samples are used. Because it scales to higher-dimensional data, we also tested Gaussian dipole smoothing on CIFAR-10 and ImageNet, where it led to modest improvements in certificates, in particular at $N = 10^6$. In Appendix C, we show the absolute, rather than relative, certified accuracy curves for the experiments shown in Figures 5 and 6. These plot show that higher-order smoothing techniques (SoS and Gaussian Dipole smoothing) are mostly beneficial for increasing the certificates of images with small certified radii. In cases where certificates are already large, increased estimation error can lead to a decrease in certificates, but this effect is small relative to the magnitudes of these certificates (typically $< 1\%$).

## 6 CONCLUSION

In this work, we explored the limits of using gradient information to improve randomized smoothing certificates. In particular, we introduced second-order smoothing certificates and showed tight and realizable upper bounds on their maximum achievable benefits. We also proposed Gaussian dipole smoothing, a novel method for robustness certification, which can improve smoothing-based robustness certificates even on large-scale data sets. This introduces a broader question for future work: what other information about the spacial distribution of classes in randomized smoothing can be efficiently used to improve robustness certificates?

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

# A    PROOFS

## A.1    SIMPLE PROOF OF THEOREM 1 FROM COHEN ET AL. (2019)

We first provide a novel, simple, and intuitive proof for first-order randomized smoothing: this will allow us to develop methods and notations for later proofs.

**Theorem.** *Let $\epsilon \sim \mathcal{N}(0, \sigma^2 I)$. For all $\mathbf{x}, \mathbf{x}'$ with $\|\mathbf{x} - \mathbf{x}'\|_2 < \rho$, and for all $f : \mathbb{R}^d \to [0, 1]$:*

$$\mathbb{E}_\epsilon[f(\mathbf{x}' + \epsilon)] \geq \Phi\left(\Phi^{-1}\left(\mathbb{E}_\epsilon[f(\mathbf{x} + \epsilon)]\right) - \frac{\rho}{\sigma}\right) \tag{10}$$

*Where $\Phi$ is the normal cdf function and $\Phi^{-1}$ is its inverse.*

*Proof.* Let $R = \|\mathbf{x} - \mathbf{x}'\|_2$. Choose our basis so that $\mathbf{x} = \mathbf{0}$ and $\mathbf{x}' = [R, 0, 0, ..., 0]^T$ (Note that by isometry, we still have $\epsilon \sim \mathcal{N}(0, \sigma^2 I)$). Then define $g : \mathbb{R} \to [0, 1]$:

$$g(z) = \mathbb{E}_{\epsilon_2,...,\epsilon_n}[f([z, \epsilon_2, ..., \epsilon_n]^T)] \tag{11}$$

Note that:

$$\begin{aligned}
\mathbb{E}_{\epsilon_1}[g(\epsilon_1)] &= \mathbb{E}_\epsilon[f(\mathbf{x} + \epsilon)] \\
\mathbb{E}_{\epsilon_1}[g(R + \epsilon_1)] &= \mathbb{E}_\epsilon[f(\mathbf{x}' + \epsilon)]
\end{aligned} \tag{12}$$

Now, in one dimension, $\epsilon_1 \sim \mathcal{N}(0, \sigma)$, and so has a pdf of $z \to \sigma^{-1}\Phi'\left(\frac{z}{\sigma}\right)$, where $\Phi'$ is the normal pdf function. By the definition of expected value:

$$\begin{aligned}
\mathbb{E}_{\epsilon_1}[g(\epsilon_1)] &= \int_{-\infty}^{\infty} g(\epsilon_1)\sigma^{-1}\Phi'\left(\frac{\epsilon_1}{\sigma}\right) d\epsilon_1 \\
\mathbb{E}_{\epsilon_1}[g(R + \epsilon_1)] &= \int_{-\infty}^{\infty} g(R + \epsilon_1)\sigma^{-1}\Phi'\left(\frac{\epsilon_1}{\sigma}\right) d\epsilon_1 = \int_{-\infty}^{\infty} g(\epsilon_1)\sigma^{-1}\Phi'\left(\frac{\epsilon_1}{\sigma} - \frac{R}{\sigma}\right) d\epsilon_1
\end{aligned} \tag{13}$$

We perform a change of integration variables, using $y = \Phi\left(\frac{\epsilon_1}{\sigma}\right)$ (and noting that $\frac{dy}{d\epsilon_1} = \sigma^{-1}\Phi'\left(\frac{\epsilon_1}{\sigma}\right)$):

$$\begin{aligned}
\mathbb{E}_{\epsilon_1}[g(\epsilon_1)] &= \int_0^1 g(\sigma\Phi^{-1}(y))\sigma^{-1}\Phi'\left(\frac{\epsilon_1}{\sigma}\right)\frac{d\epsilon_1}{dy} dy = \int_0^1 g(\sigma\Phi^{-1}(y)) dy \\
\mathbb{E}_{\epsilon_1}[g(R + \epsilon_1)] &= \int_0^1 g(\sigma\Phi^{-1}(y))\sigma^{-1}\Phi'\left(\frac{\epsilon_1}{\sigma} - \frac{R}{\sigma}\right)\frac{d\epsilon_1}{dy} dy \\
&= \int_0^1 g(\sigma\Phi^{-1}(y))\frac{\Phi'\left(\Phi^{-1}(y) - \frac{R}{\sigma}\right)}{\Phi'\left(\Phi^{-1}(y)\right)} dy
\end{aligned} \tag{14}$$

Note that:

$$\frac{\Phi'\left(\Phi^{-1}(y) - \frac{R}{\sigma}\right)}{\Phi'\left(\Phi^{-1}(y)\right)} = \frac{e^{-\frac{1}{2}\left(\Phi^{-1}(y) - \frac{R}{\sigma}\right)^2}}{e^{-\frac{1}{2}(\Phi^{-1}(y))^2}} = e^{\Phi^{-1}(y) - \frac{R^2}{2\sigma^2}} \tag{15}$$

Also, to simplify notation, define $g_\Phi : [0,1] \to [0,1]$ as $g_\Phi(y) := g(\sigma\Phi^{-1}(y))$. Then we have (combining Equations 14 and 15):

$$
\begin{aligned}
\mathbb{E}_{\epsilon_1}[g(\epsilon_1)] &= \int_0^1 g_\Phi(y)dy \\
\mathbb{E}_{\epsilon_1}[g(R+\epsilon_1)] &= \int_0^1 g_\Phi(y)e^{\Phi^{-1}(y)-\frac{R^2}{2\sigma^2}}dy
\end{aligned}
\tag{16}
$$

Fix the expectation at $\mathbf{x}$, $\mathbb{E}_{\epsilon_1}[g(\epsilon_1)]$, at a constant $C$, let us consider the function $g_\Phi$ which minimizes the expectation at $\mathbf{x}'$:

$$
\mathbb{E}_{\epsilon_1}[g(R+\epsilon_1)] \geq \min_{\substack{g_\Phi \in [0,1] \to [0,1] \\ \int_0^1 g_\Phi(y)dy=C}} \int_0^1 g_\Phi(y)e^{\Phi^{-1}(y)-\frac{R^2}{2\sigma^2}}dy
\tag{17}
$$

However, note that $e^{\Phi^{-1}(y)-\frac{R^2}{2\sigma^2}}$ increases monotonically with $y$. Then the minimum is achieved at:

$$
g_\Phi^*(y) = \begin{cases} 1 & \text{if } y \leq C \\ 0 & \text{if } y > C \end{cases}
\tag{18}
$$

In terms of the function $g(z)$, this is:

$$
g^*(z) = \begin{cases} 1 & \text{if } z \leq \sigma\Phi^{-1}(C) \\ 0 & \text{if } z > \sigma\Phi^{-1}(C) \end{cases}
\tag{19}
$$

Then we can evaluate the minimum, using the form of the integral given in Equation 13:

$$
\begin{aligned}
\mathbb{E}_{\epsilon_1}[g(R+\epsilon_1)] &\geq \int_{-\infty}^{\infty} g^*(\epsilon_1)\sigma^{-1}\Phi'\left(\frac{\epsilon_1}{\sigma}-\frac{R}{\sigma}\right)d\epsilon_1 \\
&= \int_{-\infty}^{\sigma\Phi^{-1}(C)} \sigma^{-1}\Phi'\left(\frac{\epsilon_1}{\sigma}-\frac{R}{\sigma}\right)d\epsilon_1 \\
&= \Phi\left(\frac{\sigma\Phi^{-1}(C)}{\sigma}-\frac{R}{\sigma}\right) - \Phi\left(\frac{-\infty}{\sigma}-\frac{R}{\sigma}\right) \\
&= \Phi\left(\Phi^{-1}(C)-\frac{R}{\sigma}\right)
\end{aligned}
\tag{20}
$$

By the definition of $C$ and Equation 12, this is:

$$
\mathbb{E}_\epsilon[f(\mathbf{x}'+\epsilon)] \geq \Phi\left(\Phi^{-1}\left(\mathbb{E}_\epsilon[f(\mathbf{x}+\epsilon)]\right)-\frac{R}{\sigma}\right) \geq \Phi\left(\Phi^{-1}\left(\mathbb{E}_\epsilon[f(\mathbf{x}+\epsilon)]\right)-\frac{\rho}{\sigma}\right)
\tag{21}
$$

which was to be proven. $\qquad\square$

## A.2 SECOND ORDER SMOOTHING

**Theorem 1.** *Let $\epsilon \sim \mathcal{N}(0,\sigma^2 I)$. For all $\mathbf{x}, \mathbf{x}'$ with $\|\mathbf{x}-\mathbf{x}'\|_2 < \rho$, and for all $f : \mathbb{R}^d \to [0,1]$,*

$$
\mathbb{E}_{\epsilon_1}[f(\mathbf{x}'+\epsilon)] \geq \Phi\left(\Phi^{-1}(a'+\mathbb{E}_\epsilon[f(\mathbf{x}+\epsilon)])-\frac{\rho}{\sigma}\right) - \Phi\left(\Phi^{-1}(a')-\frac{\rho}{\sigma}\right)
\tag{22}
$$

*Where $\Phi$ is the normal cdf function, $\Phi^{-1}$ is its inverse, and $a'$ is the (unique) solution to*

$$
\Phi'(\Phi^{-1}(a')) - \Phi'(\Phi^{-1}(a'+\mathbb{E}_\epsilon[f(\mathbf{x}+\epsilon)])) = -\sigma\|\nabla_\mathbf{x}\mathbb{E}_\epsilon[f(\mathbf{x}+\epsilon)]\|_2
\tag{23}
$$

*Further, for all pairs $(\mathbb{E}_\epsilon[f(\mathbf{x}+\epsilon)], \|\nabla_\mathbf{x}\mathbb{E}_\epsilon[f(\mathbf{x}+\epsilon)]\|_2)$ which are possible, there exists a base classifier $f$ and an adversarial point $\mathbf{x}'$ such that Equation 4 is an equality.*

As show by Salman et al. (2019), we have, for all $\mathbf{x}'' \in \mathbb{R}^d$:

$$
\nabla_{\mathbf{x}''}\mathbb{E}_\epsilon[f(\mathbf{x}''+\epsilon)] = \sigma^{-2}\mathbb{E}_\epsilon[\epsilon f(\mathbf{x}''+\epsilon)]
\tag{24}
$$

Under the choice of basis of the above proof (in particular, $\mathbf{x}=\mathbf{0}$), when evaluated at $\mathbf{x}$ this becomes:

$$
\nabla_\mathbf{x}\mathbb{E}_\epsilon[f(\mathbf{x}+\epsilon)] = \sigma^{-2}\mathbb{E}_\epsilon[\epsilon f(\epsilon)]
\tag{25}
$$

Let $\boldsymbol{u} := [1, 0, 0, ..., 0]^T$, and define $g$ as in the above proof. Note that:

$$
\begin{aligned}
-\|\nabla_{\mathbf{x}}\mathbb{E}_\epsilon[f(\mathbf{x}+\epsilon)]\|_2 &\leq \boldsymbol{u}\cdot\nabla_{\mathbf{x}}\mathbb{E}_\epsilon[f(\mathbf{x}+\epsilon)] \\
&= \boldsymbol{u}\cdot\sigma^{-2}\mathbb{E}_\epsilon[\epsilon f(\epsilon)] \\
&= \sigma^{-2}\mathbb{E}_\epsilon[\epsilon_1 f(\epsilon)] \\
&= \sigma^{-2}\mathbb{E}_{\epsilon_1}[\epsilon_1[\mathbb{E}_{\epsilon_2,..,\epsilon_d}f(\epsilon)]] \\
&= \sigma^{-2}\mathbb{E}_{\epsilon_1}[\epsilon_1 g(\epsilon_1)]
\end{aligned}
\tag{26}
$$

By the definition of expectation, and again using the change of integration variables $y := \Phi\left(\frac{\epsilon_1}{\sigma}\right)$,

$$
\begin{aligned}
\mathbb{E}_{\epsilon_1}[\epsilon_1 g(\epsilon_1)] &= \int_0^1 \sigma\Phi^{-1}(y)g(\sigma\Phi^{-1}(y))\sigma^{-1}\Phi'\left(\frac{\epsilon_1}{\sigma}\right)\frac{d\epsilon_1}{dy}dy \\
&= \int_0^1 \sigma\Phi^{-1}(y)g(\sigma\Phi^{-1}(y))dy
\end{aligned}
\tag{27}
$$

Define:

$$
\begin{aligned}
C &:= \mathbb{E}_{\epsilon_1}[g(\epsilon_1)] \quad (= E_\epsilon[f(\mathbf{x}+\epsilon)) \\
C' &:= \mathbb{E}_{\epsilon_1}[\epsilon_1 g(\epsilon_1)] \quad (\geq -\sigma^2\|\nabla_{\mathbf{x}}\mathbb{E}_\epsilon[f(\mathbf{x}+\epsilon)]\|_2)
\end{aligned}
\tag{28}
$$

Then, by Equations 16 and 27, and defining $g_\Phi$, as above,

$$
\mathbb{E}_{\epsilon_1}[g(R+\epsilon_1)] \geq \min_{\substack{g_\Phi\in[0,1]\to[0,1] \\ \int_0^1 g_\Phi(y)dy=C \\ \int_0^1 \sigma\Phi^{-1}(y)g_\Phi(y)dy=C'}} \int_0^1 g_\Phi(y)e^{\Phi^{-1}(y)-\frac{R^2}{2\sigma^2}}dy
\tag{29}
$$

Note that our constraints are linear in the space of functions; we can then introduce Lagrange multipliers:

$$
\begin{aligned}
\min_{g_\Phi\in[0,1]\to[0,1]} \int_0^1 g_\Phi(y)e^{\Phi^{-1}(y)-\frac{R^2}{2\sigma^2}}dy &- \lambda_1\left(\int_0^1 g_\Phi(y)dy - C\right) \\
&- \lambda_2\left(\int_0^1 \sigma\Phi^{-1}(y)g_\Phi(y)dy - C'\right)
\end{aligned}
\tag{30}
$$

$$
\min_{g_\Phi\in[0,1]\to[0,1]} \int_0^1 g_\Phi(y)e^{\Phi^{-1}(y)-\frac{R^2}{2\sigma^2}}dy - \lambda_1 g_\Phi(y) - \lambda_2\sigma\Phi^{-1}(y)g_\Phi(y)dy + \text{ constants}
\tag{31}
$$

$$
\min_{g_\Phi\in[0,1]\to[0,1]} \int_0^1 g_\Phi(y)\left(e^{\Phi^{-1}(y)-\frac{R^2}{2\sigma^2}} - \lambda_1 - \lambda_2\sigma\Phi^{-1}(y)\right)dy + \text{ constants}
$$

This is simply the inner product between $g_\Phi$ and a function: the inner product is minimized by setting $g_\Phi = 1$ where the expression is negative, and $g_\Phi = 0$ where the expression is positive:

$$
g_\Phi^*(y) = \begin{cases} 1 & \text{if } e^{\Phi^{-1}(y)-\frac{R^2}{2\sigma^2}} \leq \lambda_1 + \lambda_2\sigma\Phi^{-1}(y) \\ 0 & \text{if } e^{\Phi^{-1}(y)-\frac{R^2}{2\sigma^2}} > \lambda_1 + \lambda_2\sigma\Phi^{-1}(y) \end{cases}
\tag{32}
$$

Sign changes occur at:

$$
\Phi^{-1}(y) = -W\left(-\frac{e^{-\frac{R^2}{2\sigma^2}-\frac{\lambda_1}{\lambda_2}}}{\lambda_2}\right) - \frac{\lambda_1}{\lambda_2}
\tag{33}
$$

Where $W$ is the product-log (Lambert W) function. This returns zero, one, or two possible values, depending on the argument (zero values if the argument $< -e^{-1}$, two values on $[-e^{-1}, 0)$, and one value on non-negative arguments). Therefore there are at most two sign changes. Also, note that as $y \to 1$, $\Phi^{-1}(y) \to \infty$, taking the limit in Equation 32, we know that

$$
g_\Phi^*(1) = 0.
$$

Therefore, taking into account the constraint $\int_0^1 g_\Phi(y)dy = C$, we know that $g_\Phi^*$ is either:

- 0 everywhere (and $C = 0$), if no sign changes
- 1 at $y < C$, 0 otherwise, if one sign change
- 1 on the interval $[a, a + C]$, 0 otherwise, for some $a \in [0, 1 - C]$, if two sign changes

In fact, the final case includes the first two, so all that we need to do now is find $a$ to satisfy the $C'$ constraint. This constraint (Equation 27) becomes:

$$\int_a^{a+C} \Phi^{-1}(y) dy = \frac{C'}{\sigma} \tag{34}$$

Because $\Phi^{-1}(y)$ is monotone increasing, the LHS of Equation 34 is a monotone increasing function of $a$. Using the indefinite integral of $\Phi^{-1}$:

$$\int \Phi^{-1}(y) dy = \int \sqrt{2} \, \mathrm{erf}^{-1}(2y - 1) dy = -\frac{1}{\sqrt{2\pi}} e^{-(\mathrm{erf}^{-1}(2y-1))^2} + C = -\Phi'(\Phi^{-1}(y)) + C$$

Where $\mathrm{erf}^{-1}$ is the inverse error function. Then the constraint becomes:

$$\Phi'(\Phi^{-1}(a)) - \Phi'(\Phi^{-1}(a + C)) = \frac{C'}{\sigma} \tag{35}$$

We can now evaluate the value of the smoothed function at $\mathbf{x}'$, again using the form of the integral given in Equation 13:

$$
\begin{aligned}
\mathbb{E}_{\epsilon_1}[g(R + \epsilon_1)] &\geq \int_{-\infty}^{\infty} g^*(\epsilon_1) \sigma^{-1} \Phi'\left(\frac{\epsilon_1}{\sigma} - \frac{R}{\sigma}\right) d\epsilon_1 \\
&= \int_{\sigma\Phi^{-1}(a)}^{\sigma\Phi^{-1}(a+C)} \sigma^{-1} \Phi'\left(\frac{\epsilon_1}{\sigma} - \frac{R}{\sigma}\right) d\epsilon_1 \\
&= \Phi\left(\Phi^{-1}(a + C) - \frac{R}{\sigma}\right) - \Phi\left(\Phi^{-1}(a) - \frac{R}{\sigma}\right)
\end{aligned}
\tag{36}
$$

If we consider the form of this integral in Equation 29, which reduces to simply:

$$\int_a^{a+C} e^{\Phi^{-1}(y) - \frac{R^2}{2\sigma^2}} dy \tag{37}$$

we see that is is a monotonically increasing function in $a$. Furthermore, we have that the LHS of Equation 35 is monotonic in $a$. Therefore, if we define $a'$ as the solution to:

$$\Phi'(\Phi^{-1}(a')) - \Phi'(\Phi^{-1}(a' + C)) = -\sigma \|\nabla_{\mathbf{x}} \mathbb{E}_{\epsilon}[f(\mathbf{x} + \epsilon)]\|_2 \tag{38}$$

Then by Equation 26, we know $a' \leq a$. Then because the RHS of Equation 36 is also monotonic in $a$,

$$\mathbb{E}_{\epsilon_1}[g(R + \epsilon_1)] \geq \Phi\left(\Phi^{-1}(a' + C) - \frac{R}{\sigma}\right) - \Phi\left(\Phi^{-1}(a') - \frac{R}{\sigma}\right) \tag{39}$$

By the definitions of $g$ and $C$, and because the RHS is monotonically decreasing in $R$ (See Equation 37), we can conclude the theorem as stated.

Further, we can conclude that an equality case is possible by noting that it is achieved by the function $g^*(\mathbf{z})$ as described above: the minimal $f^*(\mathbf{z})$ can then be constructed as $f^*(z, \cdot, \cdot, ...) := g^*(z)$. Note that we also need Equation 26 to be tight: this is achieved where the adversarial direction $\mathbf{x}' - \mathbf{x}$ is parallel to the gradient of the smoothed function.

### A.3 PRACTICAL CERTIFICATION ALGORITHM

Define

$$
\begin{aligned}
\underline{C} &:= \text{lower bound on } C \\
\overline{C'} &:= \text{upper bound on } C'
\end{aligned}
\tag{40}
$$

Note that

$$
\begin{aligned}
\|\nabla_{\mathbf{x}} \mathbb{E}_{\epsilon}[f(\mathbf{x} + \epsilon)]\|_2^2 &= \\
\sigma^{-4} \mathbb{E}_{\epsilon}[\epsilon f(\epsilon)] \cdot \mathbb{E}_{\epsilon}[\epsilon f(\epsilon)] &= \\
\sigma^{-4} \mathbb{E}_{\epsilon}[\epsilon f(\epsilon)] \cdot \mathbb{E}_{\epsilon'}[\epsilon' f(\epsilon')] &= \\
\sigma^{-4} \mathbb{E}_{\epsilon, \epsilon'}[(\epsilon f(\epsilon)) \cdot (\epsilon' f(\epsilon'))]
\end{aligned}
\tag{41}
$$

**Theorem 2.** *Let* $V := \mathbb{E}_{\epsilon,\epsilon'}[(\epsilon f(\mathbf{x}+\epsilon)) \cdot (\epsilon' f(\mathbf{x}+\epsilon'))]$, *and* $\tilde{V}$ *be its empirical estimate. If* $n$ *pairs of samples* $(= N/2)$ *are used to estimate* $V$, *then, with probability at most* $\eta$, $\mathbb{E}[V] - \tilde{V} \geq t$, *where:*

$$t = \begin{cases} 4\sigma^2 \sqrt{-\frac{d}{n}\ln(\eta)} & if -2\ln(\eta) \leq dn \\ -\frac{4\sqrt{2}\sigma^2}{n}\ln(\eta) & if -2\ln(\eta) > dn \end{cases} \tag{42}$$

Wainwright (2019) gives the following condition for any *centered* (mean-zero) sub-exponential random variable $X$:

**Definition 1.** *A centered R.V.* $X$ *is (a,b)-subexponential if:*

$$\mathbb{E}[e^{\lambda X}] \leq e^{a^2\lambda^2/2}, \qquad \forall \; \lambda \in [-b^{-1}, b^{-1}] \tag{43}$$

First, we establish bounds for $\epsilon \cdot \epsilon'$. (This can be considered a simplified case of Gaussian chaos of the second order, see Vershynin (2018)). For each $i \in [d]$, $\epsilon_i$ and $\epsilon'_i$ are independent Gaussian random variables. Recall the moment-generating function for a Gaussian:

$$\mathbb{E}[e^{t\epsilon_i}] = e^{\sigma^2 t^2/2} \quad \forall \; t \tag{44}$$

Then for the product $\epsilon_i \epsilon'_i$ we have that:

$$\mathbb{E}[e^{\lambda \epsilon_i \epsilon'_i}] = \mathbb{E}_{\epsilon_i}\mathbb{E}_{\epsilon'_i}[[e^{\lambda \epsilon_i \epsilon'_i}]] = \mathbb{E}_{\epsilon_i}[e^{\epsilon_i^2(\lambda^2\sigma^2/2)}] \tag{45}$$

Note that this has a similar form to the moment generating function of the Chi-squared distribution for $k = 1$:

$$\mathbb{E}_{\epsilon_i}[e^{\epsilon_i^2 t}] = \frac{1}{\sqrt{1 - 2\sigma^2 t}} \tag{46}$$

Then:

$$\mathbb{E}[e^{\lambda \epsilon_i \epsilon'_i}] = \mathbb{E}_{\epsilon_i}[e^{\epsilon_i^2(\lambda^2\sigma^2/2)}] = \frac{1}{\sqrt{1 - \lambda^2\sigma^4}} \leq e^{\lambda^2\sigma^4} \quad \forall \lambda^2\sigma^4 \leq \frac{1}{2} \tag{47}$$

Where the final inequality can be shown by observing that, if $\lambda^2\sigma^4 \leq 1/2$:

$$\frac{1}{1 - \lambda^2\sigma^4} = 1 + \frac{\lambda^2\sigma^4}{1 - \lambda^2\sigma^4} \leq 1 + 2\lambda^2\sigma^4 \leq e^{2\lambda^2\sigma^4} \tag{48}$$

and taking square roots. Because $\epsilon_i \epsilon'_i$ is centered, this implies that $\epsilon_i \epsilon'_i$ is $(\sqrt{2}\sigma^2, \sqrt{2}\sigma^2)$-subexponential. Now, $\epsilon \cdot \epsilon'$ is simply the sum of $d$ such identical, independent, centered subexponential variables: by (Wainwright (2019) Equation 2.18), we conclude that $\epsilon \cdot \epsilon'$ is $(\sqrt{2}\sigma^2\sqrt{d}, \sqrt{2}\sigma^2)$-subexponential. This implies:

$$\mathbb{E}[e^{\lambda \epsilon \cdot \epsilon'}] \leq e^{2\sigma^4 d\lambda^2/2}, \qquad \forall \; \lambda \in [-(\sqrt{2}\sigma^2)^{-1}, (\sqrt{2}\sigma^2)^{-1}] \tag{49}$$

Recall that the quantity which we are measuring is $(\epsilon f(\epsilon)) \cdot (\epsilon' f(\epsilon'))$. For notation convenience, let $v(\epsilon, \epsilon') : \mathbb{R}^d \times \mathbb{R}^d \to \{0, 1\}$ be defined as $f(\epsilon)f(\epsilon')$, so that the quantity of interest is

$$V := \epsilon \cdot \epsilon' v(\epsilon, \epsilon') \tag{50}$$

We further define a centered version of this quantity

$$V' := V - \mathbb{E}[V] \tag{51}$$

We now introduce an important lemma:

**Lemma 1.** $V'$ *is* $(2\sqrt{2}\sigma^2\sqrt{d}, 2\sqrt{2}\sigma^2)$-*subexponential.*

*Proof.* Define

$$p := \Pr_{\epsilon,\epsilon'}[v(\epsilon, \epsilon') = 1] \tag{52}$$

Then:

$$V = \epsilon \cdot \epsilon' v(\epsilon, \epsilon') = \epsilon \cdot \epsilon' - (1 - v(\epsilon, \epsilon'))\epsilon \cdot \epsilon' \tag{53}$$

$$\mathbb{E}[V] = \mathbb{E}[\epsilon \cdot \epsilon'] - \mathbb{E}[(1 - v(\epsilon, \epsilon'))\epsilon \cdot \epsilon'] = -\mathbb{E}[(1 - v(\epsilon, \epsilon'))\epsilon \cdot \epsilon'] \tag{54}$$

$$\mathbb{E}[V] = \mathbb{E}[\epsilon \cdot \epsilon' v(\epsilon, \epsilon')] = p\mathbb{E}[\epsilon \cdot \epsilon' | v(\epsilon, \epsilon') = 1] = -(1-p)\mathbb{E}[\epsilon \cdot \epsilon' | v(\epsilon, \epsilon') = 0] \tag{55}$$

Therefore

$$
\begin{aligned}
V' &= \epsilon \cdot \epsilon' v(\epsilon, \epsilon') - \mathbb{E}[\epsilon \cdot \epsilon' v(\epsilon, \epsilon')] \\
&= \epsilon \cdot \epsilon' v(\epsilon, \epsilon') - v(\epsilon, \epsilon')\mathbb{E}[\epsilon \cdot \epsilon' v(\epsilon, \epsilon')] - (1 - v(\epsilon, \epsilon'))\mathbb{E}[\epsilon \cdot \epsilon' v(\epsilon, \epsilon')]
\end{aligned}
\tag{56}
$$

$$
\begin{aligned}
V' &= \epsilon \cdot \epsilon' v(\epsilon, \epsilon') - pv(\epsilon, \epsilon')\mathbb{E}[\epsilon \cdot \epsilon' v(\epsilon, \epsilon') | v(\epsilon, \epsilon') = 1] \\
&\quad + (1-p)(1 - v(\epsilon, \epsilon'))\mathbb{E}[\epsilon \cdot \epsilon' v(\epsilon, \epsilon') | v(\epsilon, \epsilon') = 0]
\end{aligned}
\tag{57}
$$

Define:

$$A := \epsilon \cdot \epsilon' v(\epsilon, \epsilon') + (1 - v(\epsilon, \epsilon'))\mathbb{E}[\epsilon \cdot \epsilon' | v(\epsilon, \epsilon') = 0] \tag{58}$$

$$B := -pv(\epsilon, \epsilon')\mathbb{E}[\epsilon \cdot \epsilon' v(\epsilon, \epsilon') | (\epsilon, \epsilon') = 1] - p(1 - v(\epsilon, \epsilon'))\mathbb{E}[\epsilon \cdot \epsilon' | v = 0] \tag{59}$$

$$V' = A + B \tag{60}$$

Trivially, we have:

$$\mathbb{E}[e^{\lambda \epsilon \cdot \epsilon'}] = p\mathbb{E}[e^{\lambda \epsilon \cdot \epsilon'} | v(\epsilon, \epsilon') = 1] + (1-p)\mathbb{E}[e^{\lambda \epsilon \cdot \epsilon'} | v(\epsilon, \epsilon') = 0] \tag{61}$$

However, note that also:

$$
\begin{aligned}
\mathbb{E}[e^{\lambda A}] &= p\mathbb{E}[e^{\lambda A} | v(\epsilon, \epsilon') = 1] + (1-p)\mathbb{E}[e^{\lambda A} | v(\epsilon, \epsilon') = 0] \\
\mathbb{E}[e^{\lambda A}] &= p\mathbb{E}[e^{\lambda \epsilon \cdot \epsilon'} | v(\epsilon, \epsilon') = 1] + (1-p)e^{\lambda \mathbb{E}[\epsilon \cdot \epsilon' | v(\epsilon, \epsilon') = 0]}
\end{aligned}
\tag{62}
$$

Then by Jensen's inequality, we have:

$$\mathbb{E}[e^{\lambda A}] \leq \mathbb{E}[e^{\lambda \epsilon \cdot \epsilon'}] \quad \forall \lambda \tag{63}$$

Similarly:

$$
\begin{aligned}
\mathbb{E}[e^{\lambda B}] &= p\mathbb{E}[e^{\lambda B} | v(\epsilon, \epsilon') = 1] + (1-p)\mathbb{E}[e^{\lambda B} | v(\epsilon, \epsilon') = 0] \\
\mathbb{E}[e^{\lambda B}] &= pe^{-p\lambda E[\epsilon \cdot \epsilon' | v(\epsilon, \epsilon') = 1]} + (1-p)e^{-p\lambda \mathbb{E}[\epsilon \cdot \epsilon' | v(\epsilon, \epsilon') = 0]}
\end{aligned}
\tag{64}
$$

Again, by Jensen's inequality:

$$\mathbb{E}[e^{\lambda B}] \leq \mathbb{E}[e^{-p\lambda \epsilon \cdot \epsilon'}] \quad \forall \lambda \tag{65}$$

$$\mathbb{E}[e^{\lambda B}] \leq \mathbb{E}[e^{-p\lambda \epsilon \cdot \epsilon'}] \leq e^{2\sigma^4 dp^2 \lambda^2 / 2} \leq e^{2\sigma^4 d\lambda^2 / 2}, \qquad \forall \; -p\lambda \in [-(\sqrt{2}\sigma^2)^{-1}, (\sqrt{2}\sigma^2)^{-1}] \tag{66}$$

Because $p \leq 1$, we then have:

$$\mathbb{E}[e^{\lambda B}] \leq e^{2\sigma^4 d\lambda^2 / 2}, \qquad \forall \; \lambda \in [-(\sqrt{2}\sigma^2)^{-1}, (\sqrt{2}\sigma^2)^{-1}] \tag{67}$$

In other words, we have shown that both $A$ and $B$ are both $(\sqrt{2}\sigma^2\sqrt{d}, \sqrt{2}\sigma^2)$-subexponential. Then, by Cauchy-Schwartz:

$$
\begin{aligned}
\mathbb{E}[e^{\lambda V'}] \\
&= \mathbb{E}[e^{\lambda A} e^{\lambda B}] \\
&\leq \sqrt{\mathbb{E}[e^{2\lambda A}]\mathbb{E}[e^{2\lambda B}]} \\
&\leq \sqrt{e^{8\sigma^4 d\lambda^2 / 2} e^{8\sigma^4 d\lambda^2 / 2}} \quad \forall 2\lambda \in [-(\sqrt{2}\sigma^2)^{-1}, (\sqrt{2}\sigma^2)^{-1}]
\end{aligned}
\tag{68}
$$

$$\mathbb{E}[e^{\lambda V'}] \leq e^{8\sigma^4 d\lambda^2 / 2} \quad \forall \lambda \in [-(2\sqrt{2}\sigma^2)^{-1}, (2\sqrt{2}\sigma^2)^{-1}] \tag{69}$$

In other words, $V'$ is $(2\sqrt{2}\sigma^2\sqrt{d}, 2\sqrt{2}\sigma^2)$-subexponential. $\qquad \square$

Finally, using the form of the one-sided Bernstein tail bound for subexponential random variables given in Wainwright (2019), we have, given $n$ measurements and an empirical mean estimate of $V$ as $\tilde{V}$:

$$\Pr(\mathbb{E}[V] - \tilde{V} \geq t) \leq \begin{cases} e^{\frac{-nt^2}{16d\sigma^4}} & \text{if } t \leq 2\sqrt{2}\sigma^2 d \\ e^{\frac{-nt}{4\sqrt{2}\sigma^2}} & \text{if } t > 2\sqrt{2}\sigma^2 d \end{cases} \tag{70}$$

Then, given a failure rate $\eta$, we can compute the minimum deviation $t$ such that the failure probability is less than $\eta$:

$$t = \begin{cases} 4\sigma^2 \sqrt{-\frac{d}{n}\ln(\eta)} & \text{if } -2\ln(\eta) \leq dn \\ -\frac{4\sqrt{2}\sigma^2}{n}\ln(\eta) & \text{if } -2\ln(\eta) > dn \end{cases} \tag{71}$$

## A.4   DIPOLE SMOOTHING

**Theorem 3.** *Let $\epsilon \sim \mathcal{N}(0, \sigma^2 I)$. For all $\mathbf{x}, \mathbf{x}'$ with $\|\mathbf{x} - \mathbf{x}'\|_2 < \rho$, and for all $f : \mathbb{R}^d \to [0, 1]$, define:*

$$
\begin{aligned}
C^S &:= \mathbb{E}_\epsilon[f(\mathbf{x} + \epsilon)f(\mathbf{x} - \epsilon)] \\
C^N &:= \mathbb{E}_\epsilon[f(\mathbf{x} + \epsilon) - f(\mathbf{x} + \epsilon)f(\mathbf{x} - \epsilon)]
\end{aligned}
\tag{72}
$$

*Then:*

$$
\begin{aligned}
\mathbb{E}_{\epsilon_1}[f(\mathbf{x}' + \epsilon)] \geq\ &\Phi\left(\Phi^{-1}(C^N) - \frac{\rho}{\sigma}\right) \\
&+ \Phi\left(\Phi^{-1}(\frac{1 + C^S}{2}) - \frac{\rho}{\sigma}\right) \\
&- \Phi\left(\Phi^{-1}(\frac{1 - C^S}{2}) - \frac{\rho}{\sigma}\right)
\end{aligned}
\tag{73}
$$

*Where $\Phi$ is the normal cdf function and $\Phi^{-1}$ is its inverse.*

*Proof.* As in the proof of Theorem A.1, let $R = \|\mathbf{x} - \mathbf{x}'\|_2$, and choose our basis so that $\mathbf{x} = \mathbf{0}$ and $\mathbf{x}' = [R, 0, 0, ..., 0]^T$.
First, for $f : \mathbb{R}^d \to [0, 1]$, we define a decomposition into *symmetric* and *non-symmetric* components, $f^S, f^N : \mathbb{R}^d \to [0, 1]$:

$$
\begin{aligned}
f^S(\epsilon) &:= f(\epsilon)f(-\epsilon) \\
f^N(\epsilon) &:= f(\epsilon) - f(\epsilon)f(-\epsilon)
\end{aligned}
\tag{74}
$$

Note that $f(\epsilon) = f^S(\epsilon) + f^N(\epsilon)$ and also that $f^S(\epsilon) = f^S(-\epsilon)$. Define $g^S(z), g^N(z) : \mathbb{R} \to [0, 1]$ by analogy to Equation 12. By linearity of expectation, note that $g(z) = g^S(z) + g^N(z)$. Also note that:

$$
\begin{aligned}
g^S(-z) &= \mathbb{E}_{\epsilon_2,...,\epsilon_n}[f^S([-z, \epsilon_2, ..., \epsilon_n]^T)] \\
&= \mathbb{E}_{-\epsilon_2,...,-\epsilon_n}[[f^S([-z, -\epsilon_2, ..., -\epsilon_n]^T)] \\
&= \mathbb{E}_{\epsilon_2,...,\epsilon_n}[[f^S([-z, -\epsilon_2, ..., -\epsilon_n]^T)] \\
&= \mathbb{E}_{\epsilon_2,...,\epsilon_n}[[f^S([z, \epsilon_2, ..., \epsilon_n]^T)] \\
&= g^S(z)
\end{aligned}
\tag{75}
$$

Similarly, define $g_\Phi^S$ and $g_\Phi^N$. We still have:

$$
g_\Phi(y) = g(\sigma\Phi^{-1}(y)) = g^S(\sigma\Phi^{-1}(y)) + g^N(\sigma\Phi^{-1}(y)) = g_\Phi^S(y) + g_\Phi^N(y)
\tag{76}
$$

Also (using $\Phi^{-1}(y) = -\Phi^{-1}(1 - y)$):

$$
g_\Phi^S(y) = g^S(\sigma\Phi^{-1}(y)) = g^S(-\sigma\Phi^{-1}(1 - y)) = g^S(\sigma\Phi^{-1}(1 - y)) = g_\Phi^S(1 - y)
\tag{77}
$$

Note that all of the mechanics of the proof of Theorem 1 can be applied to $f, f^S$ and $f^N$. Following Equation 13, we have:

$$
\begin{aligned}
C^S &= \mathbb{E}_\epsilon[f^S(\mathbf{x} + \epsilon)] = \int_0^1 g_\Phi^S(y)dy \\
C^N &= \mathbb{E}_\epsilon[f^N(\mathbf{x} + \epsilon)] = \int_0^1 g_\Phi^N(y)dy \\
C &:= \mathbb{E}_\epsilon[f(\mathbf{x} + \epsilon)] = \int_0^1 g_\Phi(y)dy = \int_0^1 g_\Phi^S(y) + g_\Phi^N(y)dy = C^S + C^N
\end{aligned}
\tag{78}
$$

We may then write the minimization in Equation 17, fixing $C^N$ and $C^S$ as constants separately[2]:

$$\mathbb{E}_{\epsilon_1}[g(R+\epsilon_1)] \geq \min_{\substack{g_\Phi^S, g_\Phi^N \in [0,1] \to [0,1] \\ \int_0^1 g_\Phi^S(y)dy = C^S \\ \int_0^1 g_\Phi^N(y)dy = C^N}} \int_0^1 g_\Phi(y) e^{\Phi^{-1}(y) - \frac{R^2}{2\sigma^2}} dy$$

$$= \min_{\substack{g_\Phi^S, g_\Phi^N \in [0,1] \to [0,1] \\ \int_0^1 g_\Phi^S(y)dy = C^S \\ \int_0^1 g_\Phi^N(y)dy = C^N}} \int_0^1 (g_\Phi^S(y) + g_\Phi^N(y)) e^{\Phi^{-1}(y) - \frac{R^2}{2\sigma^2}} dy \tag{79}$$

$$= \min_{\substack{g_\Phi^S \in [0,1] \to [0,1] \\ \int_0^1 g_\Phi^S(y)dy = C^S}} \int_0^1 g_\Phi^S(y) e^{\Phi^{-1}(y) - \frac{R^2}{2\sigma^2}} dy$$

$$+ \min_{\substack{g_\Phi^N \in [0,1] \to [0,1] \\ \int_0^1 g_\Phi^N(y)dy = C^N}} \int_0^1 g_\Phi^N(y) e^{\Phi^{-1}(y) - \frac{R^2}{2\sigma^2}} dy$$

The second minimum can be computed as in the proof of Theorem 1, it is simply $\Phi\left(\Phi^{-1}(C^N) - \frac{R}{\sigma}\right)$. For the first minimum, we consider the additional constraint, that $g_\Phi^S(y) = g_\Phi^S(1-y)$. Then we can rewrite the integral as:

$$\int_0^1 g_\Phi^S(y) e^{\Phi^{-1}(y) - \frac{R^2}{2\sigma^2}} dy$$

$$= \int_0^{\frac{1}{2}} g_\Phi^S(y) e^{\Phi^{-1}(y) - \frac{R^2}{2\sigma^2}} dy + \int_{\frac{1}{2}}^1 g_\Phi^S(1-y) e^{\Phi^{-1}(y) - \frac{R^2}{2\sigma^2}} dy$$

$$= \int_0^{\frac{1}{2}} g_\Phi^S(y) e^{\Phi^{-1}(y) - \frac{R^2}{2\sigma^2}} dy + \int_{\frac{1}{2}}^0 g_\Phi^S(y') e^{\Phi^{-1}(1-y') - \frac{R^2}{2\sigma^2}} (-1) dy' \tag{80}$$

$$= \int_0^{\frac{1}{2}} g_\Phi^S(y) e^{\Phi^{-1}(y) - \frac{R^2}{2\sigma^2}} dy + \int_0^{\frac{1}{2}} g_\Phi^S(y') e^{-\Phi^{-1}(y') - \frac{R^2}{2\sigma^2}} dy'$$

$$= e^{-\frac{R^2}{2\sigma^2}} \int_0^{\frac{1}{2}} g_\Phi^S(y) \left[ e^{\Phi^{-1}(y)} + e^{-\Phi^{-1}(y)} \right] dy$$

$$= 2e^{-\frac{R^2}{2\sigma^2}} \int_0^{\frac{1}{2}} g_\Phi^S(y) \cosh(\Phi^{-1}(y)) dy$$

So the minimization becomes:

$$\min_{\substack{g_\Phi^S \in [0,\frac{1}{2}] \to [0,1] \\ \int_0^{\frac{1}{2}} g_\Phi^S(y)dy = \frac{1}{2}C^S}} 2e^{-\frac{R^2}{2\sigma^2}} \int_0^{\frac{1}{2}} g_\Phi^S(y) \cosh(\Phi^{-1}(y)) dy \tag{81}$$

Note that $\cosh(\Phi^{-1}(y))$ is a monotonically *decreasing* function of $y$ on the range $[0, \frac{1}{2}]$. Then the minimum is achieved by the function:

$$g_\Phi^{S*}(y) = \begin{cases} 0 & \text{if } y < \frac{1-C^S}{2} \\ 1 & \text{if } \frac{1-C^S}{2} \leq y \leq \frac{1}{2} \end{cases} \tag{82}$$

Where the value in the domain $[\frac{1}{2}, 1]$ can be computed using $g_\Phi^{S*}(1-y) = g_\Phi^{S*}(y)$ In terms of the function $g^S(z)$, this is:

$$g^{S*}(z) = \begin{cases} 1 & \text{if } |z| \leq \sigma\Phi^{-1}(\frac{1+C^S}{2}) \\ 0 & \text{if } |z| > \sigma\Phi^{-1}(\frac{1+C^S}{2}) \end{cases} \tag{83}$$

---

[2]Note that we are not considering all applicable constraints here: in particular we are not restricting the range of $g_\Phi(z)$ to $[0, 1]$ explicitly. However, the lower bound presented here must be at least as low as the lower bound with this additional constraint, so the inequality is still valid. Also, this constraint does in fact hold in the final construction.

We can now evaluate the integral, again using the form of the integral given in Equation 13:

$$
\begin{aligned}
\mathbb{E}_{\epsilon_1}[g^S(R+\epsilon_1)] &\geq \int_{-\infty}^{\infty} g^{S*}(\epsilon_1)\sigma^{-1}\Phi'\left(\frac{\epsilon_1}{\sigma} - \frac{R}{\sigma}\right) d\epsilon_1 \\
&= \int_{-\sigma\Phi^{-1}(\frac{1+C^S}{2})}^{\sigma\Phi^{-1}(\frac{1+C^S}{2})} \sigma^{-1}\Phi'\left(\frac{\epsilon_1}{\sigma} - \frac{R}{\sigma}\right) d\epsilon_1 \\
&= \Phi\left(\Phi^{-1}(\frac{1+C^S}{2}) - \frac{R}{\sigma}\right) - \Phi\left(-\Phi^{-1}(\frac{1+C^S}{2}) - \frac{R}{\sigma}\right) \\
&= \Phi\left(\Phi^{-1}(\frac{1+C^S}{2}) - \frac{R}{\sigma}\right) - \Phi\left(\Phi^{-1}(\frac{1-C^S}{2}) - \frac{R}{\sigma}\right)
\end{aligned}
\tag{84}
$$

So, combining the $g^S$ and $g^N$ terms, we have:

$$
\begin{aligned}
\mathbb{E}_{\epsilon_1}[g(R+\epsilon_1)] &\geq \Phi\left(\Phi^{-1}(C^N) - \frac{R}{\sigma}\right) \\
&+ \Phi\left(\Phi^{-1}(\frac{1+C^S}{2}) - \frac{R}{\sigma}\right) \\
&- \Phi\left(\Phi^{-1}(\frac{1-C^S}{2}) - \frac{R}{\sigma}\right)
\end{aligned}
\tag{85}
$$

From Equation 12, and noting that the last two terms together are monotonically decreasing[3] with $R$, we complete the proof. □

# B   ADDITIONAL EXPERIMENTS

Here, we present experiments at a wider range of parameters. For all figures, images misclassified or not certified for both the baseline and the tested method are not counted: the total test set size is 1000 for MNIST, 500 for CIFAR and ImageNet with $N = 10^5$, and 100 for CIFAR and ImageNet with $N = 10^6$. For all experiments, $N_0 = 100$. Also, note that we test independently for the baseline and higher-order methods (i.e., we use different smoothing samples). This is necessary to compare fairly to dipole smoothing, where the sampling method is different; however, it does lead to some noise, especially at $N = 10^5$.

## B.1   NOISE LEVEL $\sigma$.

We see (Figures 7 and 8) that at a smaller level of noise ($\sigma = 0.12$), the effect of higher-order smoothing is diminished. This can be understood in terms of the curves in Figure 1-b: lower noise leads to more inputs with higher $p_a$, which reduces the benefit of the higher-order certificate. Conversely, higher noise increases the effects of the higher-order certificates, although it also leads to decreased total accuracy. The dipole certificate underperforms at $N = 10^5, \sigma = 0.5$: this is likely due to the increase in estimation error, which becomes significant near $p_a = 0.5$.

## B.2   DIMENSIONALITY $d$

To test second-order smoothing on a lower-dimensional dataset, we performed PCA on the $7 \times 7$ MNIST images, and classified using the top 10 principal components. ($d = 10$). Results are shown in Figures 9, 10, and 11. We see that, at $N = 10^6$, second-order smoothing has a marginal positive impact at this smaller scale.

## B.3   DIPOLE SMOOTHING ON CIFAR-10

In Figure 12, we see experiments on CIFAR-10 using dipole smoothing, for a range of $\sigma \in \{0.12, 0.25, 0.50, 1.00\}$ and $N \in \{10^5, 10^6\}$. Note that dipole smoothing appears to be beneficial even at $N = 10^5$ on CIFAR-10, at all noise levels $\geq 0.25$.

---

[3]To see this, note the form of the integral of $g_S$ equal to these terms given in Equation 80

### B.4 DIPOLE SMOOTHING ON IMAGENET

In Figure 13, we see experiments on ImageNet using dipole smoothing, for a range of $\sigma \in \{0.25, 0.50, 1.00\}$ and $N \in \{10^5, 10^6\}$. There is an anomalous result for $\sigma = 0.50$, $N = 10^5$, in that this is the only case where dipole smoothing appears to perform worse than standard smoothing. However, this turns out to be a computational artifact. At both $\sigma = 0.50$ and $\sigma = 0.25$, there are a large number of images where every smoothing sample is correctly classified, so $p_a$ is as close to 1 as the measurement bounds allow. Note that if $p_a$ truly equals 1, the certified radius is infinite, so in this domain, the reported certificate is entirely a function of the estimation error. Because dipole smoothing reduces measurement precision, these samples have somewhat smaller certified radii under dipole smoothing, especially at small $N$. However, this gap should be exactly proportional to $\sigma$. The cause of the anomaly is the fact that our code (adapted from (Cohen et al., 2019)) records each radius to three significant figures. At $\sigma = 0.5$, for an image where all noise samples are correctly classified, the ratio of the dipole smoothing radius to the standard smoothing radius is reported as $1.89/1.91 = 98.95\%$, while for $\sigma = 0.25$ it is reported as $0.947/0.953 = 99.37\%$. This explains the large number of samples with reported $> 1\%$ decrease in certificates for $\sigma = 0.50, N = 10^5$.

## C  ABSOLUTE CERTIFICATES FOR MAIN-TEXT EXPERIMENTS

In Figures 14 and 15, we show the absolute, rather than relative, values of the certificates reported in the main text, compared to the baseline first-order randomized smoothing. We see that the benefit of the proposed techniques is greatest for images with small absolute certificates, and that, on CIFAR-10 and ImageNet, there is some disadvantage to dipole smoothing on the largest possible certificates, where all smoothing samples are classified correctly. This is because, for these images, the certificate depends entirely on estimation error.

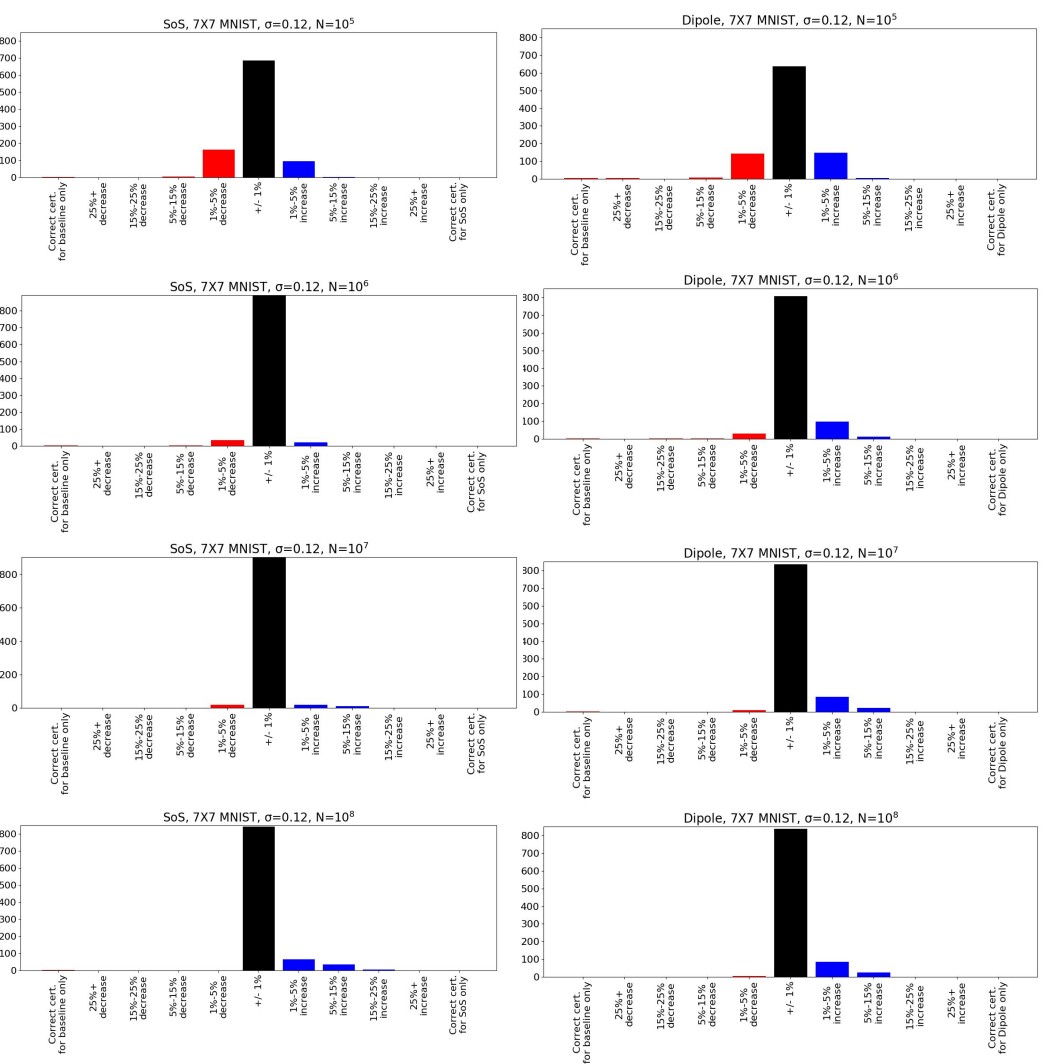

Figure 7: $7 \times 7$ MNIST, $\sigma = 0.12$

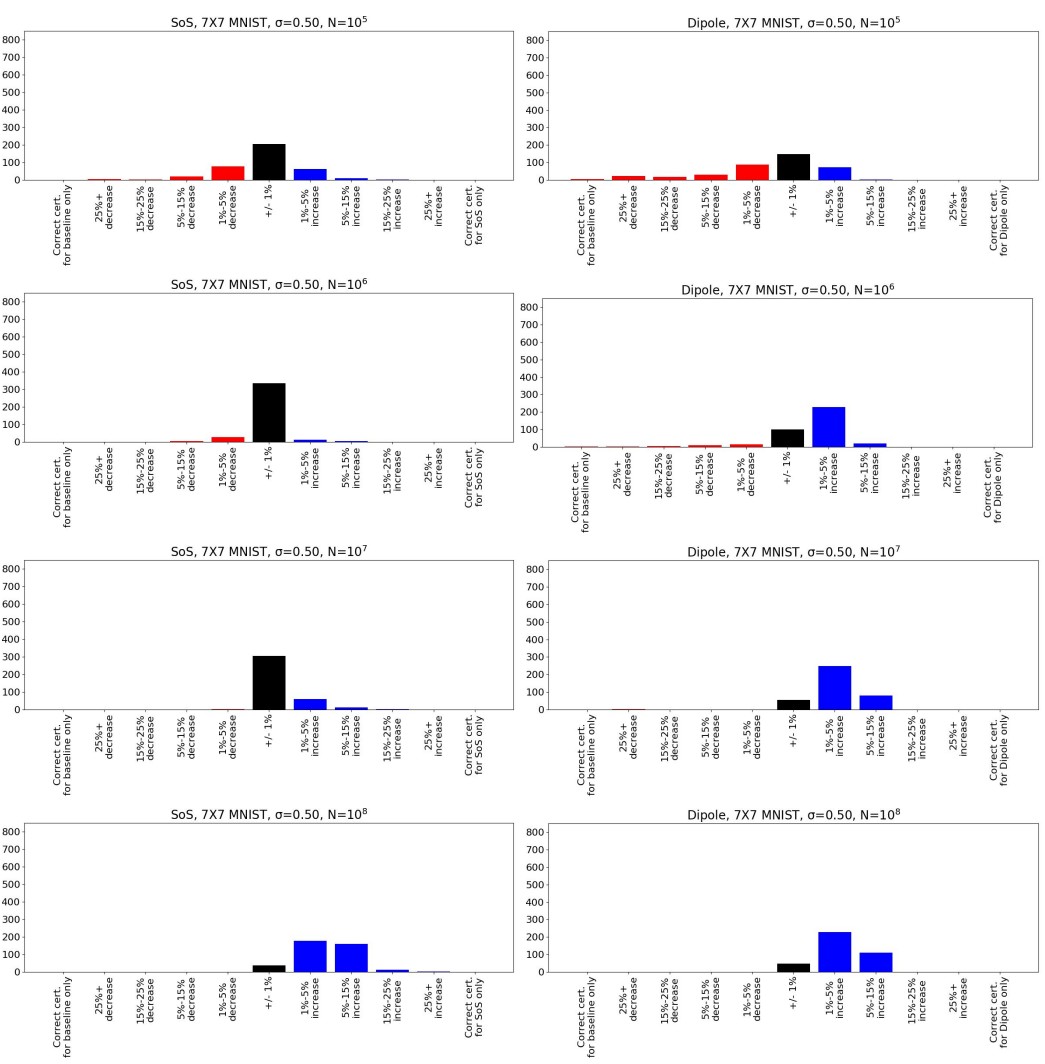

Figure 8: $7 \times 7$ MNIST, $\sigma = 0.50$

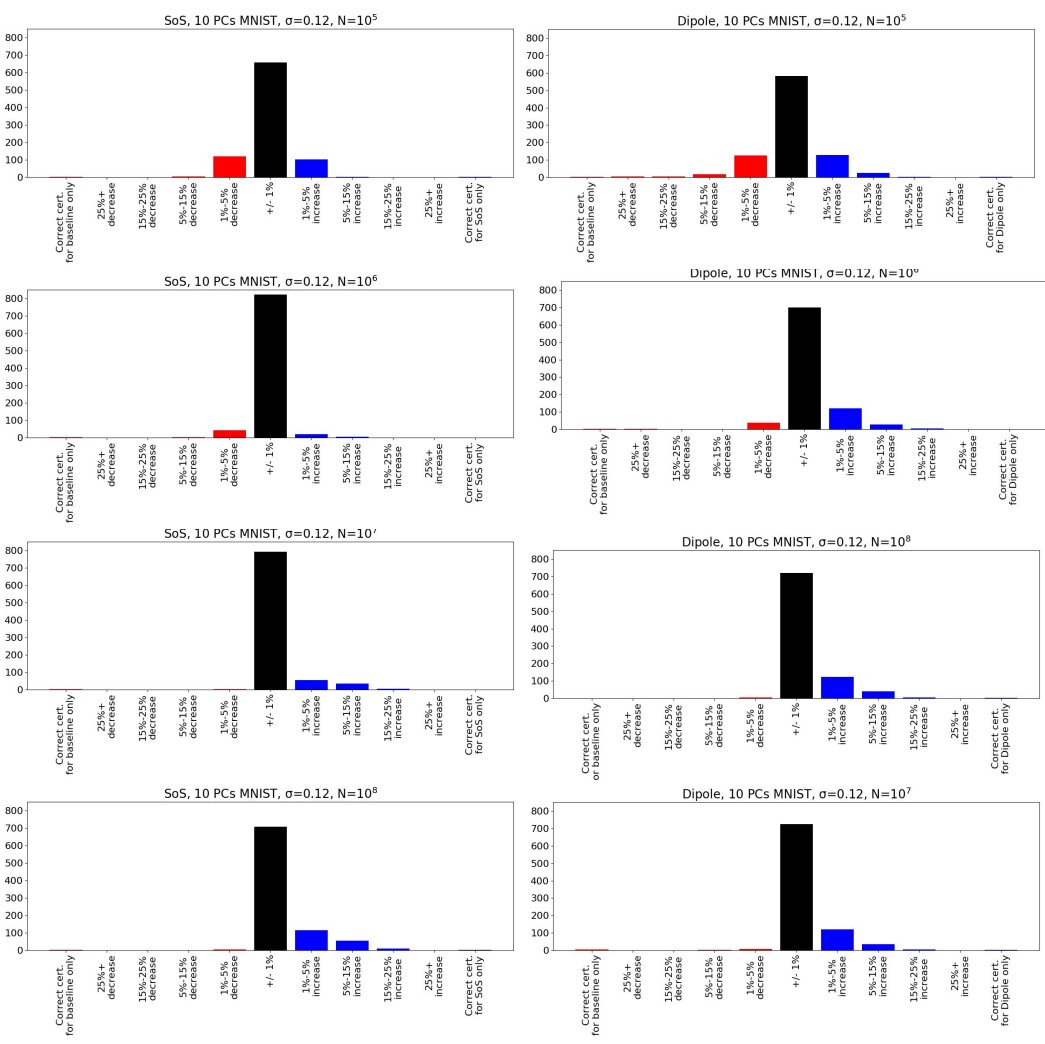

Figure 9: 10 PC MNIST, $\sigma = 0.12$

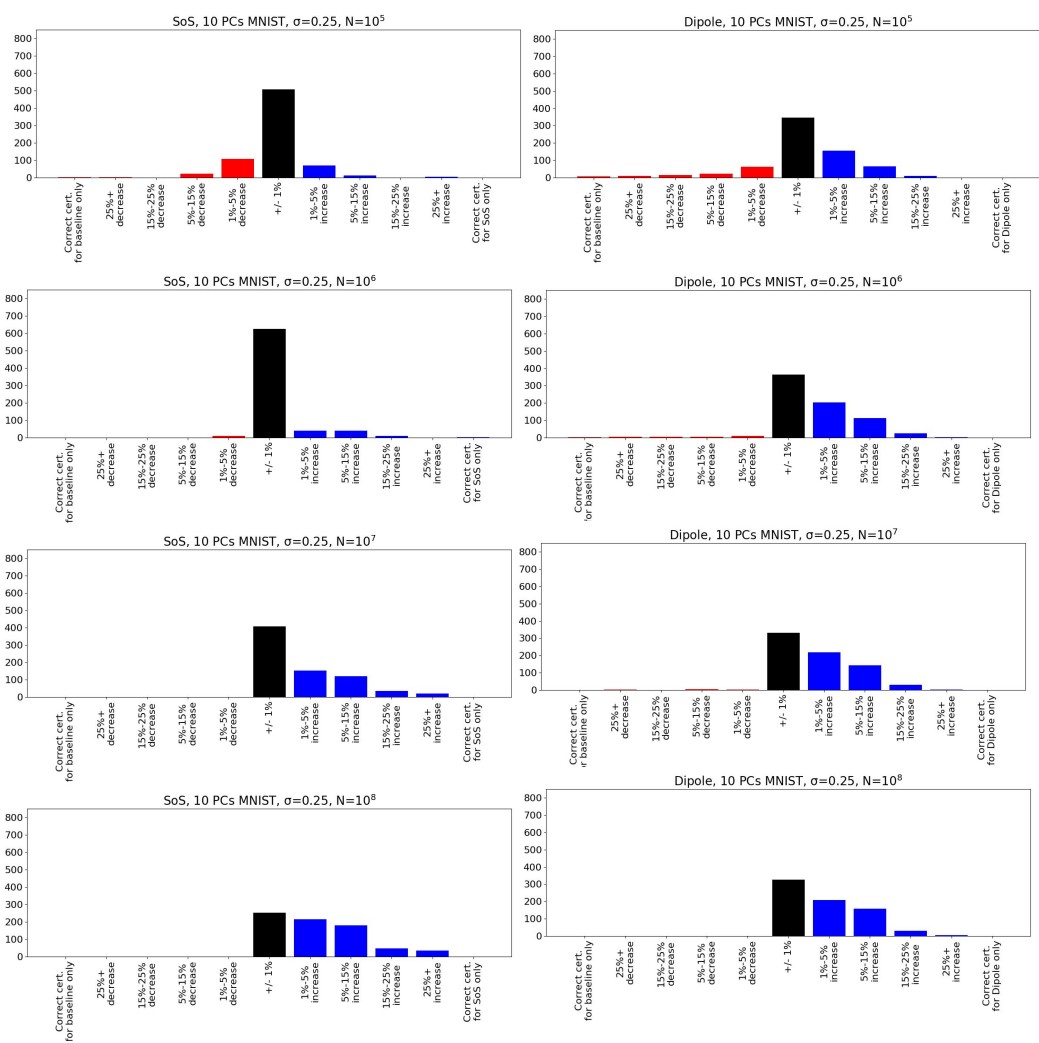

Figure 10: 10 PC MNIST, $\sigma = 0.25$

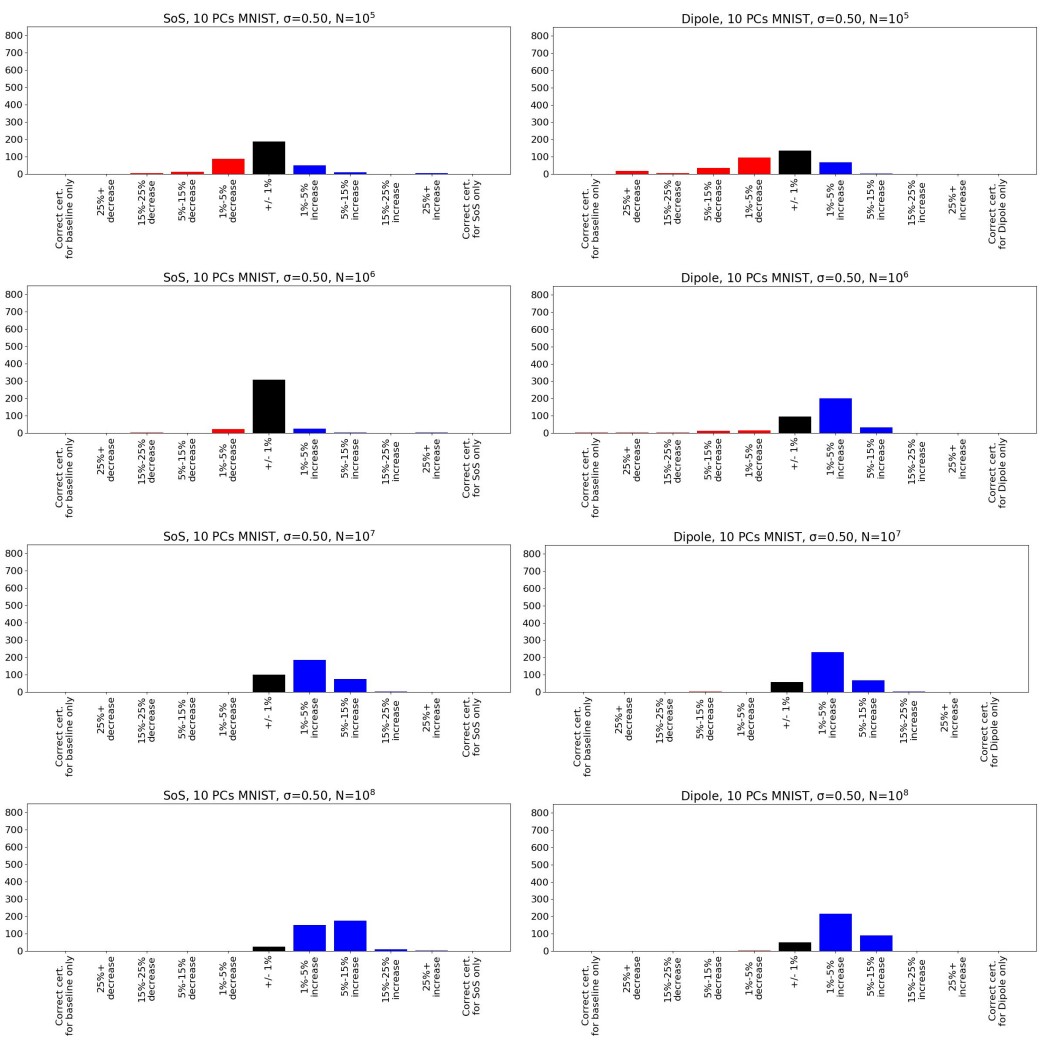

Figure 11: 10 PC MNIST, $\sigma = 0.50$

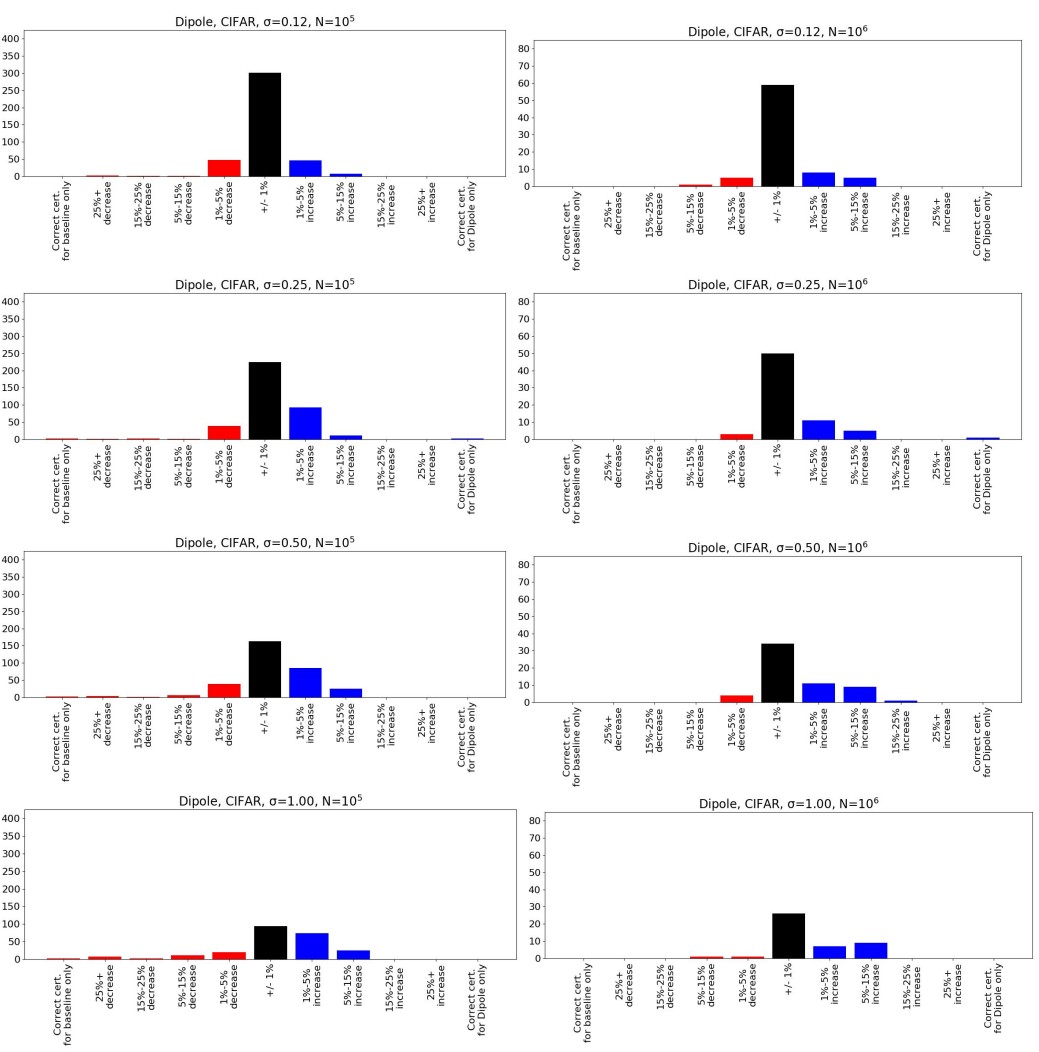

Figure 12: CIFAR-10

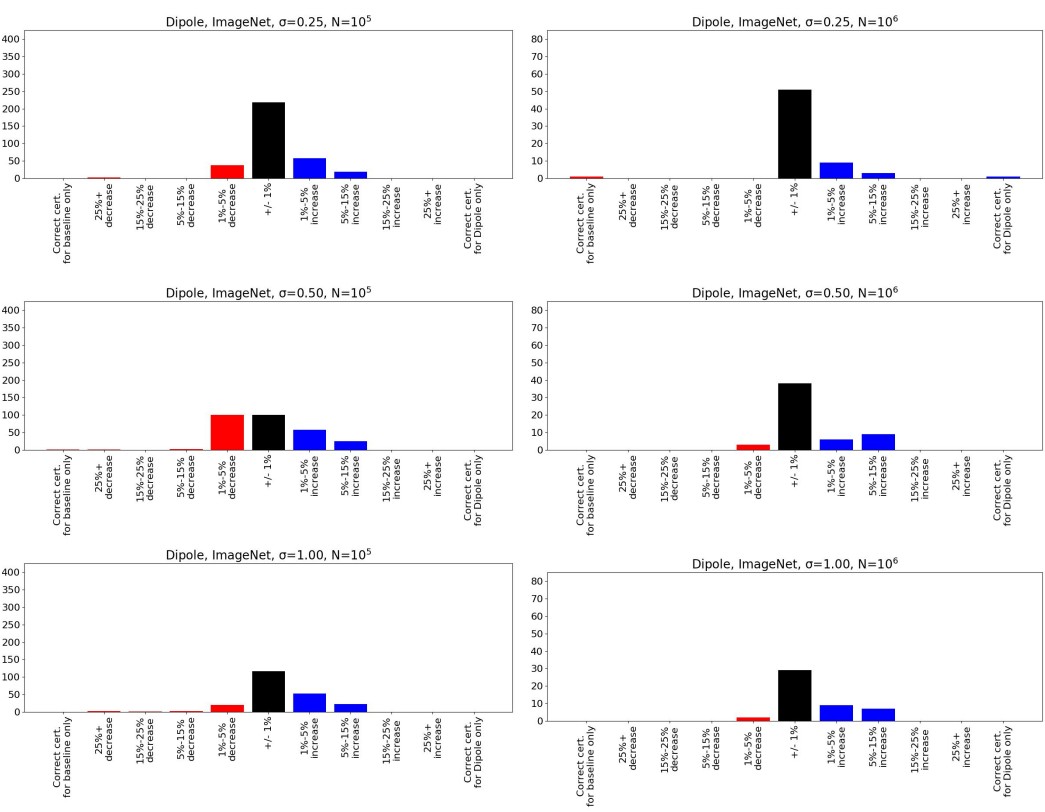

Figure 13: ImageNet

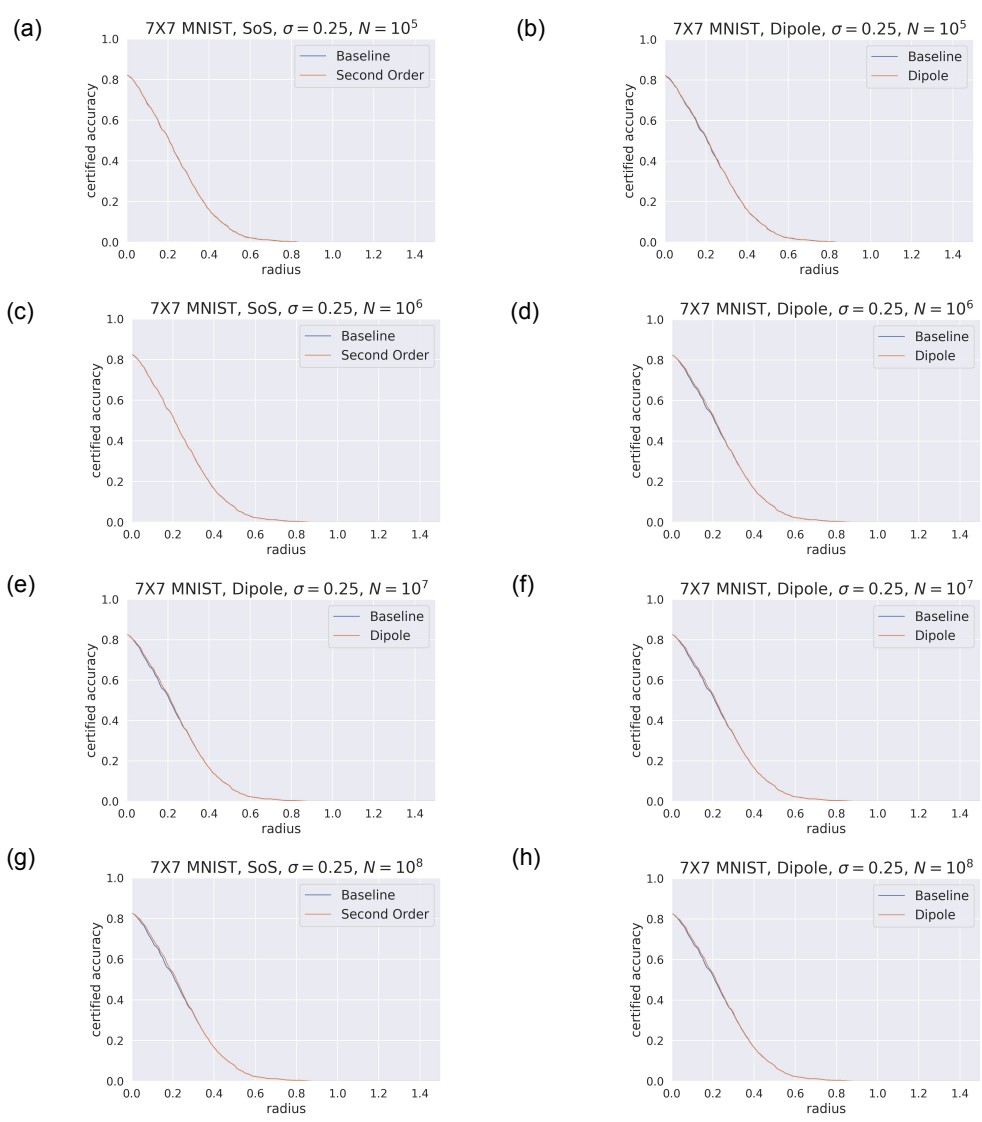

Figure 14: Experiments on $7 \times 7$ MNIST. For all, $\sigma = 0.25$. For (a, c, e, g), Second-order Smoothing is used. For (b, d, f, h), Gaussian dipole smoothing is used. For (a, b), $N = 10^5$. For (c, d), $N = 10^6$. For (e, f), $N = 10^7$. For (g, h), $N = 10^8$.

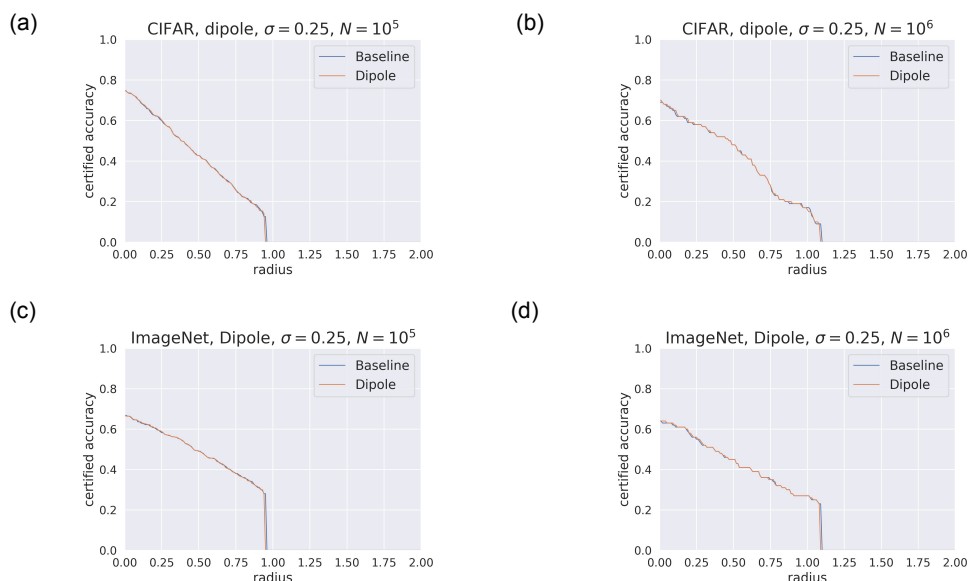

Figure 15: Dipole smoothing experiments, with $\sigma = 0.25$, on CIFAR-10 (a, b) and ImageNet (c, d). For (a, c), $N = 10^5$. For (b, d), $N = 10^6$.

