# OpenReview forum: "Tight Second-Order Certificates for Randomized Smoothing"
_ICLR.cc/2021/Conference — Reject_

### Official Review · AnonReviewer3 · 2020-10-27
**Interesting ideas; confused positioning**

**Rating:** 6
**Confidence:** 3

**Review:**

Summary:

This work extends previous results on certified robustness guarantees via randomised smoothing by incorporating gradient information of the smoothed function into the final certificate (referred to as SoS). The authors show a number of interesting properties such as (1) that the certificate is tight, by showing, for a linear classifier, the Cohen et al. [1] certificate and the SoS certificate are identical, and (2) certificates are slightly improved if gradient norms are small, but due to poor estimation of these value, sometimes the gains provided by "second-order information" are eliminated. The authors then show estimation of the gradient norm is dependent on the dimensionality of the data, and provide alternative methods that do not have a strict dependency. Although, as far as I understand, this alternative method requires computing two new lower bounds and so requires more samples to reach the same level of precision as the Cohen et al. certificate. Experiments on CIFAR-10 and ImageNet show in some cases SoS finds larger certificates.

Strength:

[+] Detailed analysis and theorem.

[+] Interesting use of geometrical information to make gradient norm estimation efficient.

[+] Experiments on large scale datasets.

Weaknesses / Questions:

1. I  am confused about the positioning of this work. The authors mention a few times that this work could be interpreted as a negative result, but the authors write in a more positive manner in other parts. Should the work be viewed as an improvement over recent work, or as a negative result that incorporating gradients doesn't improve certificates by much? If it is the latter, it isn't clear which dimension of the work is responsible for the limitations. Is it simply that, in practice, the gradient norms are substantially larger than zero, and so this new certificate more or less matches the Cohen et al. certificate? Or is it that difficulty in estimation of the gradient norms wipes away any gains? Is it both? A more substantial study of these effects would be most welcome.

2. I found it difficult to interpret the experimental results. How many of the +/- 1% changes are, in fact, decreases? It would be useful to plot  these results in terms of certified accuracy vs epsilon, as has become standard in this line of work. My concern is that if the majority of SoS certificates are marginally smaller than the Cohen et al. certificates, and a few SoS certificates are substantially higher, the overall certified accuracy curve will look worse for some epsilon values in comparison to Cohen et al..

3. Analysis does not easily extend to multi-class setting.

[1] Cohen, Jeremy M., Elan Rosenfeld, and J. Zico Kolter. "Certified adversarial robustness via randomized smoothing." arXiv preprint arXiv:1902.02918 (2019).

---

> ### Author Response · Authors · 2020-11-21
> **Reviewer #3 Rebuttal**
>
> Thank you for the comments.
>
> “I am confused about the positioning of this work. The authors mention a few times that this work could be interpreted as a negative result, but the authors write in a more positive manner in other parts. Should the work be viewed as an improvement over recent work, or as a negative result that incorporating gradients doesn't improve certificates by much?”: We show both that the maximum increase in certificates attainable using second-order information is modest (Figure 1-b) as well as that the number of samples necessary to use this information is too large to use practically (this has been improved somewhat by Mohapatra et al, as discussed above; and we have removed some language referring to the estimation method as a negative result.) However, the proposed dipole smoothing in Section 4 provides a (still modest) increase in certified robustness without requiring an asymptotic increase in the number of samples by more than a constant factor, so this method may prove practical, even if the magnitude of the improvement is relatively small.
>
> “It would be useful to plot these results in terms of certified accuracy vs epsilon, as has become standard in this line of work.”: We have included these plots in Appendix C for the data reported in the main text.
>
> “Analysis does not easily extend to multi-class setting.”: We address this in Section 3.2. Note that our method does provide _correct_ certificates in the multi-class setting, although these certificates are not necessarily tight.

---

### Official Review · AnonReviewer2 · 2020-10-30
**Merit of method has not been convincingly justified in theory and experiment**

**Rating:** 4
**Confidence:** 3

**Review:**

Summary:  This paper presents a randomized second-order smoothing certificate for providing robustness guarantees against adversarial attacks. By additionally using the gradient estimation of smoothed classifier, the proposed method has been shown to outperform the existing randomized smoothing certificate in practice. A variant of the method without explicitly estimating gradient vector has also been proposed to avoid the dependence of feature dimension in concentration analysis.

Strong points:

-S1. The addressed topic of randomized smoothing certificate is of significant importance and interest to the society of adversarial learning.

-S2. The certificate radius of the proposed method is novel as far as the reviewer knows about.

Weak points:

-W1. The advantage of the certificate radius in Theorem 1 over the existing ones is not clearly justified. Unlike the original randomized smoothing classifiers, there seems no explicit expression available for the certificate radius in Theorem 1. Based on the current bound in Equation 2, it is hard to evaluate the theoretical gain of the proposed method in robustness.

-W2. The proof of Theorem in Appendix Section A.1 looks not correct in general. The current proof argument is only customized for $x=0$ and $x’=[R,0,…,0]^\top$. It is not clear if the same claim and technique extend to arbitrary $x$ and $x’$ satisfying $\|x – x’\|\le R$.

-W3. The paper is poorly organized and presented. The main results in Section 3 are somewhat hard to follow for non-expert audiences, mainly due to the lack of a clear statement of method before indulging into theoretical analysis. Also, much of the space was allocated for elaborating a gradient estimation method which in my opinion is mostly incremental as a side contribution. Such a practice of material organization makes the paper unclear and perhaps pointless in presentation.

-W4. As an adversarial learning paper that introduces a new alternative method for robustness certificate, it is desirable to provide a sufficiently detailed experimental study in the main paper (rather than in the appendix) to more convincingly justify the real benefit of method.

-W5. There are many typos in the manuscript. Here are a few examples:

(1) Equation 4 -> Equation 2 in Theorem 1;

(2) Page 2: it possible -> it is possible;

(3) Page 5: \|nabla_x p_a(x)\| -> $\|\nabla_x p_a(x)\|$;

(4) Page 5: Salman et al. (2019) suggests -> suggest;

(5) Theorem 2: $n$ -> $N$.

---

> ### Author Response · Authors · 2020-11-21
> **Reviewer #2 Rebuttal**
>
> Thank you for the comments.
>
> “ there seems no explicit expression available for the certificate radius in Theorem 1[…] it is hard to evaluate the theoretical gain of the proposed method in robustness:” It is true that there is not a closed-form expression, however, the certified radius can still be easily computed numerically, as explained in Section 3 (at the bottom of page 3, continuing onto page 4). Moreover, the improvement over the standard randomized smoothing certificate, in terms of the two relevant variables p(x) and $|\nabla p(x)|$, is plotted in Figure 1-b.
>
> “It is not clear if the same claim and technique extend to arbitrary x and x’:” In our proofs, x and x’ are in fact arbitrary points. We are simply choosing to work in a coordinate system (with an orthonormal basis) where the origin is at x, and the first basis vector is in the direction of x’ - x. By isometry of spherical Gaussians, the form of the smoothing distribution is the same in any orthonormal basis.
>
> “much of the space was allocated for elaborating a gradient estimation method which in my opinion is mostly incremental as a side contribution:” In light of the concurrent contribution by Mohapatra et al (discussed above) we have removed some of this discussion.
>
> “experimental study in the main paper (rather than in the appendix):” We have moved some results into the main body.
>
> We have corrected the typos. However, note that n, the number of pairs of samples (= N/2, as stated) in Theorem 2 is not a typo.

---

### Official Review · AnonReviewer1 · 2020-11-06
**This paper tries to improve the randomized smoothing by leveraging the gradient information of the smoothed classifier.**

**Rating:** 5
**Confidence:** 5

**Review:**

My main concern is that the improvement of the proposed method over standard randomized smoothing is marginal.
In addition, the evaluation metric used for comparison is not standard.  It would be better to use certified accuracy as the evaluation metric used by the related works.

Pros:

1. The studied problem is important.  In particular, it's important to study the certified robustness of the classifier against adversarial perturbations.

2. The proposed second-order smoothing is novel. This paper aims to improve randomized smoothing by incorporating the gradient information of the smoothed classifier, which is novel.


Cons:
1. The evaluation metric is not standard.  Standard randomized smoothing and the follow-up work use certified accuracy as the metric to evaluate the certified robustness.  It's better for the authors to also use certified accuracy as the metric for a fair comparison.

It's not clear how different parameters (e.g., c, lambda, eta) impact the certified accuracy on CIFAR10 and ImageNet dataset.

2. The improvement of the proposed method (Gaussian dipole smoothing) over the standard randomized smoothing is marginal (most 1% in Figure 5).


Questions during the rebuttal period:

Why not using certified accuracy?

What's the impact of the parameters on the results?

Is it possible to address the issue of marginal improvement?


I would be happy to increase my score if the authors can also show results on certified accuracy and improve their results?

Typos:

1. In Theorem 1, "there exists a base classifier ... such that Equation 4 is an equality". Is it Equation 2?

2. $nabla_x$ --> $\nabla_x$.

---

> ### Author Response · Authors · 2020-11-21
> **Reviewer #1 Rebuttal**
>
> Thank you for the comments.
>
> “Standard randomized smoothing and the follow-up work use certified accuracy as the metric to evaluate the certified robustness.”: We in fact are using certified robustness: we are simply reporting the relative values to the standard certificate, rather than absolute values. This makes the difference more apparent. We have added plots of the absolute comparison to Appendix C for the data reported in the main text.
>
> “It's not clear how different parameters (e.g., c, lambda, eta) impact the certified accuracy on CIFAR10 and ImageNet dataset.”: There are not any parameters c or lambda. Could you clarify your comment? For other hyperparameters (number of samples, noise sigma), we test over a wide range of values in Appendix B.
>
> “The improvement of the proposed method (Gaussian dipole smoothing) over the standard randomized smoothing is marginal (most 1% in Figure 5).”:  This is inaccurate: indeed, for some samples, the improvement is over 5%.
>
> We have fixed the typos, thanks.

---

> > ### Comment · AnonReviewer1 · 2020-11-24
> > **No improvement on certified accucary.**
> >
> > Thank you for the response!  I checked the results in Appendix C and found that the certified accuracies of the proposed method are (almost) the same as those of the baseline across all certified radius. I would maintain my score as the proposed method did not advance the existing method in terms of certified accuracy.

---

> > > ### Author Response · Authors · 2020-11-24
> > > **Author Response**
> > >
> > > While the benefits of the proposed methods are admittedly modest (although it is inaccurate to say there is "no improvement", especially for smaller certified radii), we believe that there is still merit to our work, for two reasons:
> > >
> > > - The tightness of the certificate proposed in Theorem 1 constitutes an important negative result, as it puts an absolute upper bound on the possible benefits to using gradient information in randomized smoothing, regardless of gradient estimation method (see Figure 1-b).
> > >
> > > - The proposed Gaussian Dipole smoothing method allows improvements to certified robustness in a manner similar to using gradient information, but with no explicit dependence on dimensionality in empirical measurement. This contribution goes beyond the concurrent work (Mohapatra et al.) discussed above, where the number of samples required still scales with $\sqrt{d}$.

---

### Author Response · Authors · 2020-11-21
**General Comment about Concurrent Work**

After the submission of this paper, an independent, concurrent work, (Mohapatra et al, NeurIPS 2020) has been released which derives a bound identical to ours for second-order smoothing certificates for the $L_2$ norm. However, that work describes a sampling algorithm where the number of samples $N$ required to bound the gradient norm scales asymptotically with $\sqrt{d}$, rather than $d$. We have added a reference to this concurrent work with an explanation of this difference, and removed the discussion suggesting that the scaling with $d$ is unavoidable, as it is apparently not.

Note that our dipole smoothing method allows for sampling with *no* dependence on $d$, so our paper still represents a meaningful contribution beyond the contributions of Mohapatra et al.

Mohapatra, Jeet, Ching-Yun Ko, Tsui-Wei Weng, Pin-Yu Chen, Sijia Liu, and Luca Daniel. "Higher-Order Certification for Randomized Smoothing." Advances in Neural Information Processing Systems 33 (2020).

---

### Decision · Program_Chairs · 2021-01-07
**Final Decision**

**Decision:**

Reject

**Comment:**

The authors develop a novel robustness certificate based on randomized smoothing that accounts for second-order smoothness of functions smoothed with Gaussian noise. They develop a variant of Gaussian smoothing based on these insights that improves sample-efficiency of randomized smoothing using gradient information.

While the ideas presented were interesting, reviewers were concerned about the quality of presentation of the paper (confused positioning of results relative to prior work) as well as the lack of significant improvements upon existing methods in the experimental section. Overall, the paper is borderline based on the reviewers' comments and ratings - however, there is not sufficient evidence to justify acceptance.

I would encourage the authors to consider a significant revision to improve the clarity of contributions made and strengthen experimental results to demonstrate significant improvements, which would validate the power of the theoretical ideas presented.